

# Hydrological and Runoff Formation Processes Based on Isotope
# Tracing During Ablation Period in the Third Polar Region
Zong-Jie Li[1*], Zong-Xing Li[2*], Ling-Ling Song[3*], Jin-Zhu Ma[1*]
[1]Key Laboratory of Western China's Environmental Systems (Ministry of Education),College of
Earth Environmental Science, Lanzhou University, Lanzhou, Gansu 730000, China
[2]Key Laboratory of Ecohydrology of Inland River Basin/Gansu Qilian Mountains Ecology
Research Center, Northwest Institute of Eco-Environment and Resources, Chinese Academy of
Sciences, Lanzhou 730000, China
[3]College of Forestry, Gansu Agricultural University, Lanzhou, Gansu 730070, China
*Corresponding author: Tel: 86+18993033525, E-mail: lzjie314@163.com (Zong-Jie Li),
lizxhhs@163.com (Zong-Xing Li).



**Abstract:** This study focused on the hydrological and runoff formation processes of
river water in the source regions of the Yangtze river during different ablation
episodes in 2016 and the ablation period from 2016 to 2018. The effects of altitude
were greater for the river in the glacier permafrost area than for the mainstream and
the permafrost area during the total ablation period in 2016. There was a significant
negative correlation (at the 0.01 level) between precipitation and $\delta^{18}O$, while a
significant positive correlation was evident between precipitation and d-excess. More
interestingly, significant negative correlations appeared between $\delta^{18}O$ and temperature,
relative humidity, and evaporation. A mixed segmentation model for end-members
was used to determine the proportion of the contributions of different water sources to
the target water body. The proportions of precipitation, supra-permafrost water, and
glacier and snow meltwater for the mainstream were 41.70%, 40.88%, and 17.42%,
respectively. The proportions of precipitation, supra-permafrost water, and glacier and
snow meltwater were 33.63%, 42.21%, and 24.16% for the river in the glacier
permafrost area and 20.79%, 69.54%, and 9.67%, respectively, for that in the
permafrost area. The supra-permafrost water was relatively stable during the different
ablation periods, becoming the main source of runoff in the alpine region, except for
precipitation, during the total ablation period.
**Keywords:** River water, stable isotope, ablation period, source region, Yangtze River
**1. Introduction**



Liquid precipitation, glaciers, snow, and permafrost in cold regions are important
components of hydrological processes, serve as a key link in the water cycle, and are
amplifiers and indicators of climate change (Yang et al., 2012; Chang et al., 2015; Li
et al., 2016a; 2016b; 2018). They are not only important as the recharge sources of
water in river basins but are also important resources to support regional development
(Halder et al., 2015; Lafrenière et al., 2019). The runoff system in the source area of
the Yangtze River consists of alpine glaciers, snow, frozen soil, and liquid
precipitation. The temporal and spatial variations of runoff components are of great
significance for water levels during wet and dry years in terms of ecological
protection and the distribution of water resources (Wang et al., 2012; Pan et al., 2017;
Mu et al., 2018). Therefore, studying changes in the composition of runoff and its
hydrological effect in cold areas can not only consolidate theories on runoff research,
prediction, and adaptation, but also have important practical significance for
construction, industry, and agriculture in cold regions (Wang et al., 2009; 2017; Wang
et al., 2019).

The stable isotope tracer technique has become an important research method in
hydrology. In recent years, the response of hydrological processes to climate change
in cold regions has become a hot topic in the field of global change, which has greatly
promoted the application of the stable isotope and chemical ion tracing methods in the
analysis of runoff in cold regions (Li et al., 2015; 2019; Qu et al., 2017; Zhu et al.,
2019). Liu et al. (2004) systematically studied the contribution of glacier and snow



meltwater to runoff in a cold area in Colorado, USA. It was found that the
contribution of glacier and snow meltwater to runoff in spring was as high as 82%.
Boucher and Carey (2010) systematically studied runoff segmentation in permafrost
basins. Maurya et al. (2011) found that the average contribution of meltwater to runoff
was 32% in typical glacial basins on the southern slope of the Himalayas. The
application of the stable isotope tracer method in the analysis of runoff components in
the cold regions of China has been relatively small. Gu and Longinelli (1993) first
used $\delta^{18}O$ as a tracer in the Urumqi River in the Tianshan Mountains. The recharge
water source can be separated    into rainfall, snow meltwater, groundwater, and ice
melt water. The results showed that groundwater and snow melt water were the major
recharge sources of the Urumqi River in different periods and locations. Since then,
Kong and Pang (2012) have studied the contribution of meltwater to runoff and its
climatic sensitivity in two typical glacial basins in the Tianshan Mountains. The
composition of runoff from the Tizinafu River in the Tianshan Mountains shows that
the average contribution of snow melt water is 43% (Fan et al., 2015). The
contribution of glacier and snow meltwater to runoff in the Baishui River in the
Yulong Snow Mountains was 53.4% in summer (Pu et al., 2013). A study of the
Babao River and the Hulugou basin in the Qilian Mountains showed that different
water sources were fully mixed into groundwater before recharging rivers in this
alpine cold region, and that the contribution of meltwater in the cryosphere to runoff
in the cold region was as high as 33% (Li et al., 2014a; 2014b). Although these
studies determined the contribution of precipitation and glacier and snow meltwater to





runoff in the cold regions, they neglected the contribution of supra-permafrost water
to runoff and its impact on hydrological processes (Prasch et al., 2013; Lutz et al.,
2014). On the one hand, it increases the uncertainty of runoff analysis in the cold
regions. On the other hand, it is difficult to comprehensively evaluate the impact of
components on the runoff process and the hydrological effects in cold regions.

The source of the Yangtze River, which is a typical alpine frozen soil area, is an
important ecological barrier and a protected water source in China (Liang et al., 2008;
Li et al., 2017). The regional climate shows a significant warm and wet trend against
the background of global climate change. Regional evapotranspiration increases
because it is affected by this, and ice and snow resources exhibit an accelerating
melting trend (Kang et al., 2007; Wang et al., 2019). The ground temperature of the
permafrost increases, causing it to melt significantly. The active layer becomes thicker
and degenerates remarkably (Shi et al., 2019). Given this background, the temporal
and spatial patterns, mechanisms, and influences of precipitation, glacier and snow
meltwater, meltwater in the active layer, and groundwater in the region undergo
profound changes and impact runoff processes (Wu et al., 2015). These significant
impacts and their hydrological effects on the entire basin have gradually become
prominent.

In summary, due to the lack of data and the difficulty of observation and sampling in
cold regions, current studies have paid more attention to the study of hydrological



processes and water cycle characteristics at the watershed scale from the macroscopic
point of view. However, there is a lack of in-depth study on the mechanism of the
temporal and spatial variations of runoff components from the microscopic point of
view, and the understanding of its hydrological effects is still in the exploratory stage.
At present, although stable isotope tracer techniques have been applied to the analysis
of runoff in cold regions, most of the current studies are limited to the assessment of
the contribution and impact of glacier and snow melt water but neglect the significant
role of liquid precipitation increase and melt water in the active layer. The results in a
lack of systematic understanding of the hydrological effects of runoff composition
changes in cold regions. Meanwhile, different types of tributaries in runoff-producing
areas are the key to runoff-producing processes and are the main links to
understanding hydrological processes in cold regions. It is urgent to develop an
understanding of how runoff is produced. In addition, the current study of
hydrological processes in the source area of the Yangtze River focuses on the
variation in runoff itself and its response mechanism to climate change, lacking
in-depth analysis of runoff components and its hydrological effects. Therefore, taking
the source area of the Yangtze River as an example, we conduct a study into the
temporal and spatial variations of isotopes in different tributary rivers under the
background of climate warming and their influencing factors by using the methods of
field observation, experimental testing, stable isotope tracing, and analytical modeling
of end-element mixed runoff. Based on the conversion signals of stable isotopes in
each link of the runoff process, this study further explores the hydraulic relations,



recharge-drainage relations and their transformation paths, and the processes of each
water body, and determines the composition of runoff, quantifies the contribution of
each runoff component to different types of tributaries, and analyzes the hydrological
effects of the temporal and spatial variation of runoff components. On the one hand,
the research results can reveal the evolution mechanism of runoff in cold regions
under the background of climate warming. On the other hand, it provides parameter
support and a theoretical basis for the simulation and prediction of runoff changes in
cold regions, and then provides a scientific basis for a more systematic understanding
of the hydrological effects caused by underlying surface changes in cold regions,
ultimately    providing    decision-making    basis    for    the    rational    development    and
utilization of water resources in river basins.

**2.  Data and Methods**

**2.1  Study area**

The source region of the Yangtze River is located in the hinterland of the Tibetan
Plateau (Fig. 1). It is an important ecological barrier and water conservation region in
China. The southern boundaries are the Tanggula Mountains and Sederi Peak, which
contain the watersheds of the Nujiang River and the Lancangjiang River, respectively.
The mean altitude reaches 4000 m above sea level with a decreasing elevation from
west to east (Yu et al., 2013) that covers an area of approximately 138,000 km$^2$,
~7.8% of the total area of the Yangtze River Basin. Most tributaries start from glaciers,
and form very dense drainage networks, such as those of the Chumaer River in the



north, Tuotuohe River in the middle, and Dangqu River in the south (Pu, 1994). The
glaciers in the study area are mainly distributed along the north-oriented slopes of the
Tanggula Mountains and Sedir Mountains and the south-oriented slopes of the
Kunlun Mountains, with a total area of 1496.04 $km^2$ (Yao et al., 2014). The
permafrost has a thickness of 10 – 120 m, which accounts for 77% of the total basin
area, and most surface soils are frozen during winter and thaw in summer, and active
layer thicknesses range from 1–4 m (Gao et al., 2012). Annual average temperatures
range from 3 – 5.5°C. The annual precipitation is 221.5 – 515 mm (Yu et al., 2014).
The mean annual precipitation varies considerably over the reserve, and ~80% of the
annual precipitation occurs during summer, with the highest precipitation occurring in
August.

**2.2 Sample Collection**

This study mainly collects precipitation, glacier and snow melt-water,
supra-permafrost water and river water to systematic analysis the recharge
relationship between precipitation, glacier and snow melt-water, supra-permafrost
water and river water in the source area of the Yangtze River. The specific sampling
process is as follows:

River water: In order to analysis the spatial and temporal characteristic of stable
isotope of river water in mainstream (25 samples) and major tributary (including river
in glacier permafrost area (105 samples) and river in permafrost area (167 samples))



in the study area, All of river water samples around the traffic routes in the source area
of the Yangtze River were collected in initial ablation in 2016 (48 samples), ablation
in 2016 (88 samples), end ablation in 2016 (45 samples), ablation in 2017 (55 samples)
and ablation in 2018 (61 samples) (Fig.1).

Glacier and snow melt-water: This paper researched the hydrochemistry characteristic
of melt-water in Cryosphere (Yuzhu peak Glacier, Geladandong Glacier and
Dongkemadi Glacier) through collected water samples by fixed-point sampling from
June to September in 2016 and 2017. The samples were collected once every 10 days
at the glacier front during the ablation period. The sampling time is at 14 o'clock per
day. The sampling location is in hydrological section at the end of the glacier.

Supra-permafrost water: Supra-permafrost water is the most widely distributed
groundwater type in the SRYR, and it is mainly stored in the permafrost active layer
(Li et al.,2018). The hydrochemistry characteristic of supra-permafrost water in the
study area this paper collected water samples by comprehensive sampling from June
to September in 2016 and 2018. The sampling process is manual operation. At first, a
2-m deep profile of the permafrost active layer was dug at each of the sampling points.
Then, the collection of the water samples are immediately filtered with 0.45 um
Millipore filtration membrane. Then, samples were poured the filtered into a clean
polyethylene bottle.



Precipitation: precipitation samples were collected at Zhimenda Hydrological Station
(ZMD) at the mountain pass of the source area of the Yangtze River, Qumalai
Meteorological Station(QML) in the middle reaches of the source area and Tuotuo
River Meteorological Station(TTH) in the upper reaches of the source area. The
sampling period extended from April 1, 2016 to October 31, 2018.

Before analysis, all samples were stored at 4ºC in a refrigerator without evaporation.
Precipitation and surface water samples were analyzed for $\delta^{18}O$ and $\delta D$ by means of
laser absorption spectroscopy (liquid water isotope analyzer, Los Gatos Research
DEL-100, USA) at the Key Laboratory of Ecohydrology of Inland River Basin,
Northwest Institute of Eco-Environment and Resources, CAS. The results are reported
relative to the Vienna Standard Mean Ocean Water (VSMOW). Measurement
precisions for $\delta^{18}O$ and $\delta D$ were better than 0.5‰ and 0.2‰, respectively. Field
measurements included pH, dissolved oxygen (DO), electrical conductivity (EC), and
water temperature.

**2.3 EMMA**

Hooper (2003) introduced the end-member mixing analysis (EMMA) using
chemical/isotopic compositions in waters. The techniques involve graphical analyses,
in which chemical and isotopic parameters are used to represent the designated end
members. Tracer concentrations are constant in space and time. Essentially, the
composition of the water changing can be considered as a result of intersections



during its passage through each landscape zone. Tracers can be used to determine both
sources and flow paths. The EMMA tracer approach has been a common method for
analyzing potential water sources contributing to stream flow ( Li et al, 2014a; 2016a).
Here in a three end-member mass-balance mixing model is employed to calculate the
contribution of up to three water sources in stream water, such as the following:
$$X_S = F_1 X_1 + F_2 X_2 + F_3 X_3 \quad (1a)$$
$$Y_S = F_1 Y_1 + F_2 Y_2 + F_3 Y_3 \quad (1b)$$
In Eq. (1), X and Y represent concentrations of two types of different tracers. In this
study, $\delta^{18}O$ and deuterium excess were chosen for comparison. The subscripts
represents stream water sample, and 1, 2, and 3 represent water from the respective
contribution of three respective source waters (end members) to stream water. The
fraction of each end-member is denoted by F. The solutions for $F_1$, $F_2$, and $F_3$ in
regards to tracer concentrations in Eq. (1) can be given as:
$$F_1 = [(X_3-X_S)/(X_3-X_2)-(Y_3-Y_S)/(Y_3-Y_2)]/[(Y_1-Y_3)/(Y_3-Y_2)-(X_1-X_3)/(X_3-X_2)] \quad (2a)$$
$$F_2 = [(X_3-X_S)/(X_3-X_1)-(Y_3-Y_S)/(Y_3-Y_1)]/[(Y_2-Y_3)/(Y_3-Y_1)-(X_2-X_3)/(X_3-X_1)] \quad (2b)$$
$$F_3 = 1 - F_1 - F_2 \quad (2c)$$
This method has been used by previous study ( Li et al.,2014b; 2015; 2016b). This
study also used this method to evaluate the contribution of possible sources to the
river water.

**2.4 Uncertainty in hydrograph separation**

The uncertainty of tracer‑based hydrograph separations can be calculated using the
error propagation technique (Genereux, 1998; Klaus & McDonnell, 2013). This
approach considers errors of all separation equation variables. Assuming that the





contribution of a specific streamflow component to streamflow is a function of several
variables c1, c2, ⋯, cn and the uncertainty in each variable is independent of the
uncertainty in the others, the uncertainty in the target variable (e.g.,the contribution of
a specific streamflow component) is estimatedusing the following equation (Genereux,
1998; Uhlenbrook & Hoeg,2003):
$$W_{fx} = \sqrt{\left(\frac{\partial z}{\partial c_1} W_{c_1}\right)^2 + \left(\frac{\partial z}{\partial c_2} W_{c_2}\right)^2 + \cdots + \left(\frac{\partial z}{\partial c_n} W_{c_n}\right)^2},$$    (3)
where W represents the uncertainty in the variable specified in the ubscript. fx is the
contribution of a specific streamflow component x to streamflow. The software
package MATLAB is used to apply equation 3 to the different hydrograph separations
in this study.

**3. Results**

**3.1 Temporal Variation**

As shown in Fig. 2, there was significant difference in $\delta^{18}O$ and d-excess in the
different ablation events in 2016 and total ablation from 2016 to 2018 for the different
types of runoff. For the mainstream, the order of $\delta^{18}O$ for the different ablation
periods was initial ablation (−10.31‰) > final ablation (−12.22‰) > total ablation
(−13.51‰), while the order of $\delta^{18}O$ in ablation from 2016 to 2018 was total ablation
in 2018 (−11.21‰) > total ablation in 2017 (−13.20‰) > total ablation in 2016
(−13.51‰). The order of d-excess for the different ablation periods and total ablation



from 2016 to 2018 was: total ablation (13.57‰) > initial ablation (12.71‰) > final
ablation (12.35‰) and total ablation in 2017 (14.62‰) > total ablation in 2016
(13.57‰) > total ablation in 2018 (10.81‰) (Fig. 2a, d). For the river in the glacier
permafrost area, the order of $\delta^{18}O$ for the different ablation periods and the total
ablation from 2016 to 2018 was the same as the mainstream order, but the values of
$\delta^{18}O$ were different for the mainstream. The $\delta^{18}O$ values for the initial ablation in
2016, total ablation in 2016, final ablation in 2016, total ablation in 2017, and total
ablation in 2018 were −9.92‰, −13.29‰, −10.82‰, −12.38‰, and −11.04‰,
respectively. The order of d-excess for the different ablation periods and total ablation
from 2016 to 2018 was: total ablation (14.24‰) > initial ablation (13.02‰) > final
ablation (10.58‰) and total ablation in 2016 (14.24‰) > total ablation in 2017
(12.40‰) > total ablation in 2018 (10.49‰) (Fig. 2b, e). For the river in the
permafrost area, the order of $\delta^{18}O$ for the different ablation periods and ablation from
2016 to 2018 was: initial ablation (−10.02‰) > final ablation (−11.65‰) > total
Ablation (−12.53‰) and total ablation in 2018 (−11.17‰) > total ablation in 2017
(−11.99‰) > total ablation in 2016 (−12.53‰). This was the same as for the
mainstream and the river in the glacier permafrost area. However, the order of
d-excess for the river in the permafrost area was different than that for the river in the
glacier permafrost area. This order, for the different ablation periods and ablation
from 2016 to 2018, was as follows: final ablation (13.61‰) > total ablation
(12.25‰) > initial ablation (9.97‰), and total ablation in 2017 (13.57‰) > total
ablation in 2016 (12.25‰) > total ablation in 2018 (9.72‰) (Fig. 2c, f). In general,





the $\delta^{18}$O in the mainstream was more negative than those in the rivers in the glacier
permafrost and permafrost areas. These results may be due to the fact that the highest
runoff was for the mainstream and that the effects of dilution result in lower isotope
values. However, the $\delta^{18}$O in the river in the glacier permafrost area was more
positive than those in the mainstream and the river in the permafrost area. The effect
of evaporation could explain these results and the change in d-excess could also
demonstrate the same.

**3.2  Spatial Variation**

To analyze the spatial variation of $\delta^{18}$O based on the different ablation periods in 2016
and total ablation from 2016 to 2018, spatial interpolation of all river water samples in
the study area was performed using ArcGIS. The results are shown in Fig. 3. The $\delta^{18}$O
value in the north-central region of the study area was more positive than those in
other regions. In the southeastern part of the study area, especially the QML, ZMD,
and Tanggula Mountains, the values were more negative during the initial ablation
period. The area of positive ablation during the total ablation period, which was
concentrated mainly in the northeast part of the study area, was larger than that during
the initial ablation. The other regions, except some areas in the southwest, turned
positive. The area of positive ablation was largest during the final of the different
ablation periods in 2016; all areas, except some in the eastern region of the study area,
were positive (Fig. 3). The area of positive ablation in the central and northern regions
began to expand in 2017 compared to the area of total ablation in 2016. Furthermore,





the area of negative ablation appears mainly in the southeastern and southwestern
portions of the study area. However, the positive ablation area was also concentrated
in the central and northern regions in 2018 and it was greater than it was in 2016 and
2017. Meanwhile, the negative ablation area appeared mainly in the southeastern and
southwestern portions of the study area, but it was smaller than in 2016 and 2017.
These results may be related to evaporation, possible recharge sources, or
meteorological factors. These results were comprehensive and influenced by
meteorological factors and the type and proportion of recharge sources. The
evaporation effect was strong in the central and northern regions, which were also the
major glacier and permafrost regions. The southeastern region was the downstream
area where all runoff converged; thus, the dilution effect led to a more negative $\delta^{18}O$
here. Moreover, the Tanggula Mountains, with altitudes higher than those in other
regions, were located southwest of the study area; thus, evaporation had a low
influence on this region and the oxygen stable isotopes were more negative.

Just as with the spatial distribution of $\delta^{18}O$, there was a significant spatial distribution
of d-excess in the study area (Fig. 4). Compared to the spatial distribution of $\delta^{18}O$, the
d-excess in the central and northern regions were lower than those in the other regions.
However, d-excess was higher in the latter, especially in the southwestern regions and
in the southeastern regions during the initial ablation period. The lower area begin to
expand during the total ablation period in 2016, while the central and northeastern
regions and the Tanggula Mountains were greater. Meanwhile, the negative ablation



area continued to expand during the final ablation period; ablation was greater only in
the southeastern part of the study area. However, all regions exhibited high ablation,
especially in the Tanggula Mountains, except for areas in the eastern region where the
ablation was low during the ablation period in 2017. Moreover, the lower ablation
regions appeared mainly in the central and southeastern regions of the study area;
values were higher in the other regions, especially in the Tanggula Mountains and the
northeast. The spatial distribution of d-excess also confirmed the spatial distribution
of the oxygen stable isotope because evaporation resulted in the enrichment of
isotopes and led to a reduction in d-excess.

In general, the influence of evaporation on the isotope and d-excess was only
manifested in some places, such as the central and northern parts of the study area, in
the initial ablation and the total ablation periods. However, the influence of
evaporation on the isotope and d-excess was manifested in most places, except the
southeast of the study area. Meanwhile, these results also indicated that there may be
a hysteresis for the influence of meteorological factors on isotopes and d-excess. On
the one hand, river water was the result of the final convergence of various recharge
sources that include precipitation, supra-permafrost water, and glacier and snow
meltwater. On the other hand, meteorological factors directly affected the main
recharge sources of river water.

As shown in Fig. 6, there was a significant difference in the variation of $\delta^{18}O$ and



d-excess with altitude for the mainstream, the river in the glacier permafrost area, and
the river in the permafrost area of the study area. For the mainstream, the oxygen
stable isotope showed a decreasing trend, with increases in altitude, during the
ablation periods in 2016 and 2018. In other words, the altitude effect only appeared in
the total ablation periods during these two years and had values of −0.16‰/100 m
($p < 0.05$) and −0.14‰/100 m ($p < 0.05$), respectively. However, $\delta^{18}O$ showed an
increasing trend with an increase in altitude during the initial and final ablation
periods in 2016 and total ablation period in 2017. The anti-altitude effects of the
initial and final ablation periods in 2016, and total ablation period in 2017, were
0.11‰/100 m ($p < 0.05$), 0.13‰/100 m ($p < 0.01$), and 0.04‰/100 m ($p < 0.05$),
respectively. d-excess showed a decreasing trend during the initial and final ablation
periods in 2016 and a significant increasing trend in the total ablation period from
2016 to 2018. For the river in the glacier permafrost area, $\delta^{18}O$ showed a decreasing
trend with increase in altitude during the total ablation periods in 2016 and 2018, but
the ablation in 2018 was not significant. The altitude effect was −0.66‰/100 m
($p < 0.05$) and −0.15‰/100 m ($p > 0.05$), respectively, during the former two periods.
Moreover, a significant anti-altitude effect of 0.47‰/100 m ($p < 0.05$), 0.67‰/100 m
($p < 0.05$), and 0.97‰/100 m ($p < 0.05$), appeared in the initial and final ablation
periods in 2016 and total ablation period in 2017, respectively. Just as with the
mainstream, d-excess showed a decreasing trend in the initial and final ablation
periods in 2016 and an increasing trend in the total ablation from 2016 to 2018. For
the river in the permafrost area, $\delta^{18}O$ showed a decreasing trend with an increase in





altitude in the initial ablation period and total ablation period in 2016, with an altitude
effect of −0.38‰/100 m ($p < 0.05$) and −0.12‰/100 m ($p > 0.05$), respectively.
However, $\delta^{18}O$ showed an increasing trend with increase in altitude in the final
ablation period in 2016 and the total ablation periods in 2017 and 2018, with an
anti-altitude effect of 0.21‰/100 m ($p < 0.05$), 0.01‰/100 m ($p > 0.05$), and
0.68‰/100 m ($p < 0.05$), respectively. d-excess showed an increasing trend with
increase in altitude in the initial and final ablation periods in 2016 and total ablation
periods in 2016 and 2017. However, d-excess also showed a decreasing trend with
increase in altitude in the total ablation period in 2018.

In summary, the altitude effect mainly appeared during ablation, whether it was in the
mainstream, the river in the glacier permafrost area, or the river in the permafrost area.
The altitude effects were higher for the river in the glacier permafrost area than for the
mainstream or the river in the permafrost area during the ablation period in 2016.
Meanwhile, the anti-altitude effect of the river in the glacier permafrost area was
higher than that of the other areas. The $\delta^{18}O$ during the initial and final ablation
periods in 2016 showed a significant anti-altitude effect for the mainstream and the
river in the glacier permafrost area; a significant altitude effect appeared during the
initial ablation period for the river in the permafrost area. These results may be due to
the comprehensive influence of possible recharge sources and different recharge
proportions caused by the influence of meteorological factors.



### 3.3 Evaporation Line

The variations in the location of the evaporation line for river water during the different ablation periods in 2016 and the total ablation periods from 2016 to 2018 are shown in Fig. 6. The slope and intercept of the LEL for river water showed an increasing trend from the initial to final ablation periods in 2016. The LEL in the initial ablation period was $\delta D = 6.59\delta^{18}O - 3.60$ ($p < 0.01$) and it was $\delta D = 6.88\delta^{18}O - 1.37$ ($p < 0.01$) during the total ablation period. The LEL during the final ablation period was $\delta D = 7.39\delta^{18}O + 5.88$ ($p < 0.01$). These results indicate that the effect of evaporation on the stable isotopes in river water gradually weakened from the initial ablation to the final ablation periods. The slope and intercept of the LEL of river water during the total ablation period in 2017 were lower than those in 2016. The LEL during the total ablation period in 2017 was $\delta D = 6.59\delta^{18}O - 3.63$ ($p < 0.01$). However, whether the slope or the intercept of LEL of river water in 2018 was higher than that in 2016 and 2017, with the LEL was: $\delta D = 7.63\delta^{18}O + 5.82$ ($p < 0.01$). This phenomenon showed that the influence of evaporation on stable isotope levels was greatest during the total ablation period in 2017, followed by that in 2016. In general, the lower slope and intercept indicate that the water body was affected by evaporation or non-equilibrium dynamic fractionation. This conclusion could also explain the results of this study.

### 3.4 Recharge Sources



The distributions of δD and δ¹⁸O for river water in the different types of water, among
supra-permafrost water, glacier snow meltwater, and precipitation, are shown in Fig. 8
the different ablation periods in 2016 and ablation from 2016 to 2018. The results of
the distribution of δD and δ¹⁸O of river water indicate the possible recharge sources of
river water. However, the δD and δ¹⁸O of river water, supra-permafrost water, glacier
snow meltwater, and precipitation exhibited little change during the initial ablation in
2016 (Fig. 7a, b). This phenomenon suggests that precipitation may be the major
recharge sources for river water during the initial ablation. A plot of δD versus δ¹⁸O
for river and supra-permafrost water, glacier snow meltwater, and precipitation is
shown in Fig. 8c. The δD and δ¹⁸O values of glacier and snow meltwater from above
the LMWL are the most negative compared to other water bodies. The stable isotope
of supra-permafrost water was relatively more positive, located below the LMWL,
confirming the influence of strong evaporation. The stable isotope of river water was
close to the LMWL, and its concentration value was between precipitation, glacier
and snow meltwater, and supra-permafrost water, reflecting that river water was
recharged and affected by multi-source water in the study area. Moreover, the
distribution of river water, glacier and snow meltwater, and supra-permafrost water
also indicated that there was a hydraulic relationship between the source and target in
the different ablation periods in 2016 and ablation from 2016 to 2018.

The mixed segmentation model of the end-member is used to determine the
contribution proportions of different water sources to the target water. Owing to the



two stable isotope concentrations in different water bodies have significant spatial and
temporal differences, which can effectively distinguish different water bodies and
their mixing relationships. The d-excess and $\delta^{18}O$ are used as tracers of the mixed
segmentation model of the end-elements. As shown in Fig. 8, according to the
locations of the different types of water (mainstream, glacier permafrost area river,
and permafrost area river) and the distance from other water bodies (precipitation,
glacier and snow meltwater, and supra-permafrost water), which reflected the mixed
recharge of three water bodies, supra-permafrost water was the first end element,
precipitation was the second end element, and glacier and snow meltwater was the
third end element in the initial ablation in 2016. However, the possible recharge
sources of the mainstream, the glacier permafrost area river, and the permafrost area
river were different (Fig. 8), as the different runoffs likely have different recharge
sources and different recharge proportions. Overall, the source of the permafrost area
river was mainly the supra-permafrost water, with similar levels of precipitation in the
different periods of ablation in 2016 and the total ablation from 2016 to 2018. The
permafrost area river had the least contribution from the glacier and snow meltwater,
indicating that the supra-permafrost water was the major recharge source for the
permafrost area river followed by precipitation, and the recharge proportions also
exhibited the same trend. As the source of the glacier permafrost area river was the
same as the permafrost area river, the permafrost area river was dominated by the
supra-permafrost water, followed by precipitation and then glacier and snow
meltwater. However, the glacier permafrost area river comprised glacier and snow



meltwater more so in the total ablation period than in other periods. Compared with
the permafrost area river and the glacier permafrost area river, the mainstream was
governed by the supra-permafrost water in the initial ablation period while containing
nearly equal proportions of supra-permafrost water and precipitation in the final
ablation period. However, the mainstream received significant contributions from all
three end members in the total ablation period from 2016 to 2018 and particularly in

487 2017.


The recharge proportions of precipitation, supra-permafrost water, and glacier and
snow meltwater at different altitudes are depicted in Fig. 9, from the mixed
segmentation model of the three end-members during the ablation periods mentioned
above. The recharge proportions of the three end members in the ablation periods
were significantly different. This may be due to the different effects of the runoff
recharge sources in different ablation periods, as well as the significant differences in
recharge and drainage relationships in the different ablation periods. The recharge
proportions of precipitation in the initial ablation in 2016, total ablation in 2016, final
ablation in 2016, total ablation in 2017, and total ablation in 2018, obtained by
calculating the average contribution proportion from each altitude, were 28.71%,
44.41%, 44.60%, 42.53%, and 51.03%, respectively. Meanwhile, the recharge
proportions of supra-permafrost water in the initial ablation in 2016, total ablation in
2016, final ablation in 2016, total ablation in 2017, and total ablation in 2018 were
55.38%, 36.51%, 40.21%, 37.56%, and 28.87%, respectively. The recharge



proportions of glacier and snow meltwater in the initial ablation in 2016, total ablation
in 2016, final ablation in 2016, total ablation in 2017, and total ablation in 2018 were
15.91%, 19.08%, 15.19%, 19.90%, and 20.09%, respectively. The recharge proportion
of precipitation decreased with increase in altitude in the initial ablation, while the
proportion of supra-permafrost water and glacier and snow meltwater exhibited an
increasing trend with increase in altitude. However, the recharge proportion of the
supra-permafrost water was higher than that of precipitation or glacier and snow
meltwater, and also showed a decreasing trend from low to high altitude in the final
ablation in 2016. The proportion of glacier and snow meltwater increased with
increase in altitude, but the recharge proportion of supra-permafrost water was stable
with the change in altitude in the final ablation in 2016. The trend of precipitation and
glacier and snow meltwater for the total ablation was the same as that for the initial
and final ablation. However, the recharge proportion of precipitation was higher than
the proportion of supra-permafrost water and glacier and snow meltwater in the
ablation period. Meanwhile, the recharge proportion of glacier and snow meltwater in
ablation was higher than that in the initial and final ablation period. In general, the
recharge of supra-permafrost water to runoff was stable, whether in the different
ablation periods in 2016 or the total ablation from 2016 to 2018. However, the
proportion of supra-permafrost water was relatively low, mainly due to the larger
runoff during the ablation period.

Using the approach shown in Equation (3), the uncertainty originating from the
variation in the tracers of components and measurement methods could be calculated



separately (Uhlenbrook & Hoeg, 2003; Pu et al., 2013). According to the calculations
made using Equation (3), the uncertainty was estimated to be 0.07 for the three‐
component mixing model in the study region. The uncertainty terms for
supra-permafrost water accounted for more than 50.0% of the total uncertainty,
indicating that the $\delta^{18}O$ and $\delta D$ variations of supra-permafrost water accounted for the
majority of the uncertainty. Although there is some uncertainty for hydrograph
separation, isotope-based hydrograph separations are still valuable tools for evaluating
the contribution of meltwater to water resources, and they are particularly helpful for
improving our understanding of hydrological processes in cold regions, where there is
a lack of observational data.

**4. Discussions**

**4.1 Meteorological Factors**

To further explain the reason for the variation in temporal and spatial characteristics
of stable isotopes and LEL, this study includes the analysis of the monthly change in
precipitation, temperature, relative humidity, and evaporation during the sampling
period (from January 2016 to December 2018). The results are shown in Fig. 10. The
average of the precipitation was 371.9 mm during the sampling period, and the
precipitation in the total ablation period accounted for 78.87%. The average of the
temperature, relative humidity, and evaporation during the sampling period were
−1.42 °C, 52.20%, and 4.14 mm, respectively. However, the average of the
temperature, relative humidity, and evaporation during the total ablation period were



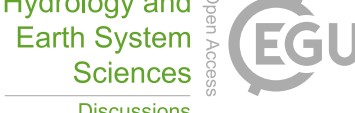

8.04 °C, 66.47%, and 5.57 mm, respectively.

More importantly, the precipitation during the initial, total, and final ablation periods
in 2016, and the total ablation periods in 2017 and 2018, were 50.40 mm, 107.90 mm,
42.90 mm, 70.60 mm, and 119.00 mm, respectively. For precipitation, the isotope
levels tend to decrease with the increase in rainfall; Precipitation is also the major
source of water for all water bodies (Maurya et al., 2011; Pu et al., 2013; Li et al.,
2014b; 2015; 2016a; 2018; Pan et al., 2017) and, in general, more precipitation
resulted in a greater dilution effect. A more negative $\delta^{18}O$ appeared in the total
ablation period in 2016 whether in all three study areas given the change in $\delta^{18}O$ (Fig.
2). This result showed that dilution does not only play an important role in the
precipitation effect; it also affects river water. However, the dilution effect was also
significant when precipitation was the major recharge source for river water
(Abongwa and Atekwana, 2018; Li et al., 2015).

Temperature for the initial, total, and final ablation periods in 2016, and the total
ablation periods in 2017 and 2018, were 6.82 °C, 9.58 °C, 3.77 °C, 9.47 °C, and
11.09 °C, respectively. For atmospheric precipitation, the lower the temperature was,
the higher the condensation degree of water vapor exhibited and the lower the isotope
content in precipitation. Therefore, there is a positive correlation between the stable
isotope and temperature in precipitation (Li et al., 2016a). However, the influence of
temperature on the stable isotope of river water was not significant from the variation





in river water isotope during the different ablation periods. However, the variation
trend of the stable isotope of river water in the total ablation period from 2016 to 2018
was similar to that for the change in temperature. Meanwhile, the variation trend of
d-excess can also be confirmed by this analysis (Fig. 2).

Relative humidity in the initial ablation, total ablation, and final ablation periods in
2016 and the total ablation periods in 2017 and 2018 were 60.07%, 63.16%, 70.57%,
63.39%, and 63.48%, respectively. When the relative humidity is low, the dynamic
fractionation increases and the slope decreases, and vice versa. The variation trend of
the slope of the LEL for the different ablation periods in 2016 was the same as that for
the change in relative humidity (Fig. 6). Meanwhile, the intercept of the LEL for the
different ablation periods in 2016 also showed the same trend.

Evaporation in the initial ablation, total ablation, and final ablation periods in 2016
and total ablation periods in 2017 and 2018 were 6.69 mm, 6.96 mm, 4.02 mm,
6.48 mm, and 6.02 mm, respectively. The stable isotopes of hydrogen and oxygen in
river water are comprehensively affected by the evaporation process, runoff change,
precipitation recharge, glacier and snow meltwater recharge, supra-permafrost water,
and evaporation loss in cold regions. During the process of evaporation, lighter water
isotopes are separated preferentially from the surface of water while heavier isotopes
are enriched in the remaining water body. Evaporation enriches the oxygen and
hydrogen stable isotopes and reduces excess deuterium (Li et al., 2015; 2018). The





trend in the oxygen isotope in the total ablation periods from 2016 to 2018 was the
same as that for the change in evaporation (Fig. 2). Meanwhile, the spatial distribution
of δ¹⁸O and d-excess also responded to this change (Fig. 3, 4).

To further analyze the influence of meteorological factors on the stable isotope, the
correlation analysis between meteorological factors and the monthly value of δ¹⁸O
and d-excess, which showed continuous observations at two fixed-point stations was
analyzed (Table 1), and the results are shown in Table 1. There was a significant
negative correlation between precipitation and δ¹⁸O at the 0.01 level (2-tailed), while
a significant positive correlation between precipitation and d-excess was also present.
More interestingly, just as with precipitation, a significant negative correlation
appeared between δ¹⁸O and temperature, relative humidity, and evaporation, with
coefficients of −0.671, −0.555, and −0.636, respectively. Meanwhile, a significant
positive correlation occurred between d-excess and temperature, relative humidity,
and evaporation, with coefficients of 0.602, 0.524, and 0.533, respectively. This
results indicated that the direct influence of meteorological factors on stable isotopes
of river water was significant and definite.

Hydrogen and oxygen isotope compositions in river water are the result of the
combined effects of the isotopes making up present in precipitation, glacier and snow
meltwater, and supra-permafrost water as well as evaporative fractionation (Li et al.,
2015). The main influential hydrometeorological factors include precipitation,




temperature, relative humidity, and evaporation. On the whole, river water isotopes
were not influenced by a single factor; instead, they were based on the comprehensive
influence of many factors in the cold regions. The influence of meteorological factors
on different types of river water (mainstream, rivers in glacier permafrost areas, and
rivers in permafrost areas) showed that apart from their directly influences, each
factor indirectly affected the river water recharge source. This indirect influence was
mainly felt on precipitation, glacier, snow, and permafrost.


**4.2 Hydrological processes**

To systematically quantify the main recharge sources of different types of runoff in
the alpine regions, the possible sources and recharge proportions of runoff of different
types in different ablation periods were deeply analyzed by using the mixed
segmentation model of the three end-members in this study. The conceptual model
map of the recharge form and proportion of the river water in the different ablation
periods is shown in Fig. 11.

For the river in the glacier permafrost area, there was a significant difference in the
recharge proportion in the runoff area, in which there were several glaciers and
permafrost in the basin, and other areas during the various ablation periods. The
proportion of recharge from precipitation during the initial, total, and final ablations in
2016, the total ablation in 2017, and the total ablation in 2018 were 27.69%, 33.71%,



639 32.38%, 33.21%, and 41.48%, respectively. However, the proportion of

640 supra-permafrost water in the initial, total, and final ablations in 2016, the total

641 ablation in 2017, and the total ablation in 2018 were 54.68%, 35.96%, 32.38%,

642 33.21%, and 41.48%, respectively. The proportions of glacier and snow meltwater in

643 the initial, total, and final ablations in 2016, the total ablation in 2017, and the total

644 ablation in 2018 were 17.63%, 30.33%, 21.24%, 29.39%, and 22.19%, respectively.

645 These results show that supra-permafrost water was the important recharge source for

646 runoff during the initial and final ablation periods. The proportion of supra-permafrost

647 water was 50.53% during the initial and final ablation periods. It was also the next

648 highest source of runoff recharge, next to precipitation, during the ablation from 2016

649 to 2018; the proportions were 36.13% and 36.66%, respectively. The recharge

650 proportions for glacier and snow meltwater was higher during the total ablation period

651 than in the initial and final ablation periods, at 19.44% and 27.30%, respectively.

652

653 For permafrost area river, the runoff area only with permafrost and no glacier in the

654 basin, there was also an obvious difference for the recharge proportion in different

655 ablation period. Compared with the glacier permafrost area river the recharge

656 proportion of supra-permafrost water was higher for permafrost area river than that

657 for the glacier permafrost area river (42.21%). The recharge proportion of

658 supra-permafrost water was 69.54%. With the same as the glacier permafrost area

659 river, the supra-permafrost water was the important recharge sources to runoff in the

660 initial and final ablation, and the proportion was 80.97% in the initial and final





ablation period. Meanwhile, the proportion of supra-permafrost water was 61.92% in
the total ablation period. The proportion was higher than that for precipitation
(24.13%) in the total ablation period. In general, the supra-permafrost water was the
major recharge source for the permafrost area river in the different ablation periods in
the study area. Meanwhile, the glacier and snow meltwater had little contribution to
the permafrost area river in the initial and final ablation periods.

For the mainstream, the recharge proportions for precipitation during the initial, total,
and final ablations in 2016, the total ablation in 2017, and the total ablation in 2018
were 28.67%, 48.35%, 43.18%, 46.97%, and 41.33%, respectively. The proportion
was 35.93% in the initial and final ablation periods and 45.55% in the total ablation
period. However, the proportions of supra-permafrost water during the initial, total,
and final ablation in 2016, the total ablation in 2017, and the total ablation in 2018
were 52.37%, 33.52%, 42.61%, 39.68%, and 38.21%, respectively. The proportion
was 47.49% during the initial and final ablation periods and 36.47% during the total
ablation period. These results indicate that, for the study area, the supra-permafrost
water was the major recharge source for the mainstream in the first two of these
ablation periods while precipitation was the major recharge source for the mainstream
in the total ablation period. The proportions of glacier and snow meltwater during the
initial, total, and final ablation in 2016, the total ablation in 2017, and the total
ablation in 2018 were 18.96%, 20.13%, 14.21%, 13.35%, and 20.46%, respectively.
The proportion of glacier and snow meltwater for the mainstream (16.59%) was





higher than that for the river in the permafrost area (3.25%) but lower than that for the
river in the glacier permafrost area (19.44%) during the initial and final ablation
periods. The former proportion was also higher than that for the river in the
permafrost area (17.98% vs 13.95%) but lower than that for the river in the glacier
permafrost area (27.30%) during the total ablation period.

The hydrological process in cold regions has one particularity. The low permeability
in permafrost layer and the freeze-thaw depths of the soil reduces soil infiltration (Wu
et al., 2015; Wang et al., 2019). Therefore, the rapid replenishment of meltwater by
runoff results in a difference in the runoff generation mechanism in the permafrost
and non-permafrost regions (Yang et al., 2010; Li et al., 2018). Moreover, because the
freeze-thaw depths of the soil changes with annual fluctuations in temperature, there
is an effect on soil water storage capacity that results in a difference in the runoff
generation mechanism during different ablation periods (Wang et al., 2019). Wang et
al. (2008) also found that the seasonal distributions and variations in rainfall runoff in
the permafrost basin were controlled by the freeze-thaw process because of the
impermeable nature of the freeze-thaw front and permafrost layer. During the initial
ablation period, the supra-permafrost water—whether in the mainstream, the river in
the glacier permafrost area, or the river in the permafrost area—was the major
recharge source. During the total ablation period, precipitation was the main source of
runoff recharge, followed by supra-permafrost water. Although there was little
difference the proportion of precipitation and supra-permafrost water during the



ablations from 2016 to 2018, precipitation was the major recharge source of runoff in
this period. Supra-permafrost water was the main source of runoff recharge in the
final ablation period, just as it was in the initial ablation period. In summary, runoff in
the cold region during the different ablation periods was mainly composed of runoff
from rainfall, meltwater, and supra-permafrost. Because of the inherent seasonal
variation in precipitation, there were significant changes in precipitation during the
different ablation periods and strong ablation periods in different years. Glacier and
snow meltwater was also greatly affected by climatic factors during the different
ablation periods, while the supra-permafrost water was relatively stable; the latter
became the main source of runoff supply, except for precipitation, in the alpine region.
Thus, with the changes that the low temperatures made in the physical properties of
the underlying surface, the change in the permafrost had the most significant effect on
the hydrological process in cold regions.

**4.3 Hydrological significance of permafrost**

The Qinghai-Tibet Plateau (QTP) is the only mid-latitude region in the world that
contains permafrost (Zhang et al., 2003). Its permafrost region is located at the source
of two major rivers (the Yangtze River and the Yellow River) in China (Yu et al., 2013;
2014). Just like the rivers in the Arctic region of Eurasia, they play an important
hydrological role in ensuring freshwater recharge and maintaining the ecological
security of the basin (Yao et al., 2014; Li et al., 2017; Wang et al., 2019). Permafrost
is an objective geological entity developed through the exchange of material and





energy between the earth and the atmosphere under the influence of the regional
geographic environment, geological structure, lithology, hydrology, and surface
characteristics given geologic history and the impact of climate change (Chang et al.,
2015). It has its own unique law of evolution and is extremely sensitive to
environmental change. The active layer of permafrost is a near-surface soil and rock
layer that thaws in the summer and freezes in winter (Wang et al., 2008; 2009; 2017;
Chang et al., 2015; Li et al., 2018). Permafrost and active layers are the main factors
controlling the hydrometeorological changes of the underlying surface, and the
freeze-thaw process of the permafrost active layer is the most important factor
affecting the process of runoff. The special water and heat exchange in the active
layer of permafrost is the key factor to maintaining the stability of the alpine
ecosystem. Permafrost, alpine marsh wetland, and alpine meadow ecosystems have
remarkable water conservation functions. They are important factors in stabilizing the
water cycle and river runoff in river source areas and have a very important impact on
regional ecology and water resource security (Yang et al., 2010; Wu et al., 2015).
Under global climate change conditions, permafrost degradation is mainly seen in
terms of changes in the active layer. In recent decades, the thickness of permafrost
active layers have changed significantly in the Qinghai-Tibet Plateau; since 1980, it
has increased by 0.71 cm/a in the eastern part of this region (Zhao et al., 2004). Jin et
al. (2006) believe that permafrost in the Qinghai-Tibet Plateau is deteriorating over a
large area because of climate change. The observed permafrost data also show a
significant increase in the thickness of the active layer in the Qinghai-Tibet Plateau

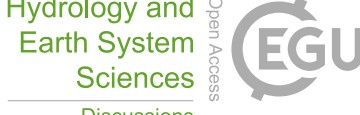
over the past 10 years.

The permafrost active layer, particularly the hydrothermal environment of the active
layer, is the most active and dominant influencing factor at the interface of the
ecological environments in cold regions (Yang et al., 2010; Wu et al., 2015; Li et al.,
2018). The change in the active layer not only changes the soil water retention
capacity, directly affecting the living environment of vegetation, but also changes the
soil freeze-thaw process in the active layer. At the same time, the energy-water
exchanges accompanied by the freeze-thaw process directly affect the redistribution
of soil water and the change in soil water capacity, the movement of water to surface
of the frozen soil, and the exchange of latent heat of the water phase transformation.
Permafrost reduces the hydraulic conductivity of soil, resulting in the reduction of
snow meltwater or precipitation infiltration, changing runoff generation, confluence
processes, and characteristics in cold regions (Boucher et al., 2010; Li et al., 2014a;
2016b; Mu et al., 2018; Shi et al., 2019).

Permafrost is the main component of ecosystem in the source area of the Yangtze
River. The source of the Yangtze River is in one of the main permafrost districts in the
Qinghai-Tibet Plateau. The permafrost area in the study area accounts for 70% of this
area (Yao et al., 2014; Li et al., 2017).The change in and distribution of permafrost
regions have a significant impact on vegetation and wetlands in this area, as the
former is one of the most sensitive to global climate change. The increase in



temperature leads to an increase in soil temperature, which deepens the active layer
significantly, and causes the permafrost to begin to degenerate. This will certainly
lead to significant changes in the ecology and water cycle of the region (McGuire et
al., 2002; Walker et al., 2003; Yang et al., 2010).

In brief, the freeze-thaw of soil in the active layer plays an important role in
controlling river runoff. The increase in melting depth leads to a decrease in the direct
runoff rate and slow dewatering process. The two processes of runoff retreat are the
result of soil freeze-thaw in the active layer. Permafrost has two hydrological
functions: on the one hand, permafrost is an impervious layer, and it has the function
of preventing surface water or liquid water from infiltrating into deep soil; on the
other hand, it forms a soil temperature gradient, which makes the soil moisture close
to the ice cover. Therefore, changes in the soil water capacity, soil water permeability,
and soil water conductivity, as well as the redistribution of water in the soil profile,
are caused by the freeze-thaw of the active layer. The seasonal freeze-thaw process of
the active layer directly leads to seasonal flow changes in surface water and
groundwater, which affects surface runoff. Climate warming is the main driving force
in the degradation of cold ecosystems (Wang et al., 2009; Wu et al., 2015; Li et al.,
2018; Wang et al., 2019).

**5. Conclusions**

Through systematically analysis of the characteristics of $\delta^{18}$O, $\delta$D, and d-excess of



river water in the different ablation periods in 2016 and the total ablation periods from
2016 to 2018, the results were as follows.
The temporal and spatial characteristics of stable isotopes of river water were
significant in the study area. The mean of $\delta^{18}O$ in TTH was −10.59‰, and the mean
of d-excess was 9.24‰, while the mean of $\delta^{18}O$ and d-excess in ZMD was −11.99‰
and 9.66‰, respectively. The oxygen isotope in ZMD was more negative than TTH,
while the d-excess in ZMD was more positive than TTH. The $\delta^{18}O$ in mainstream was
more negative than that in the glacier permafrost area river and permafrost area river.
The influence of evaporation on isotope and d-excess is only prevalent in some places,
such as the central and northern parts of the study area in the initial ablation and total
ablation periods. However, the influence of evaporation on isotope and d-excess is
prevalent in most places except the southeastern part of the study area. Meanwhile,
this results also indicated that there may be a hysteresis for the influence of
meteorological factors on isotopes and d-excess. The altitude effect is only present
during the total ablation periods in 2016 and 2018, and the altitude effect was
−0.16‰/100 m ($p < 0.05$) and −0.14‰/100 m ($p < 0.05$). The altitude effects were
higher for the glacier permafrost area river than those for the mainstream and
permafrost in the total ablation period in 2016. Meanwhile, the anti-altitude effect of
the glacier permafrost area river was higher than that of the mainstream and
permafrost area river. The $\delta^{18}O$ in the initial and final ablation periods in 2016
showed a significant anti-altitude effect for the mainstream and the glacier permafrost
area river, while a significant altitude effect appeared in the initial ablation period for



the permafrost area river. The slope of LEL for river water showed an increasing trend
from initial ablation to final ablation in 2016. Meanwhile, the intercept of LEL for
river water also increased from the initial ablation to the final ablation period.

Moreover, the average of precipitation was 371.9 mm during the sampling period, and
the precipitation in the total ablation period accounted for 78.87%. The average of the
temperature, relative humidity, and evaporation during the sampling period were
−1.42 °C, 52.20%, and 4.14 mm, respectively. However, the average of the
temperature, relative humidity, and evaporation in the ablation period were 8.04 °C,
66.47%, and 5.57 mm, respectively. Through correlation analysis, it is concluded that:
there was a significant negative correlation between the precipitation and $\delta^{18}O$ at the
0.01 level (2-tailed), while a significant positive correlation between precipitation and
d-excess. More interestingly, just as with precipitation, significant negative
correlations were prevalent between $\delta^{18}O$ and temperature, relative humidity, and
evaporation, with coefficients of −0.671, −0.555, and −0.636, respectively.

Finally, the mixed segmentation model of the end-member is used to determine the
contribution proportion of different water sources to the target water. The results
showed that the recharge proportion of precipitation decreased with an increase in
altitude in the initial ablation, while the proportions of supra-permafrost water and
glacier and snow meltwater showed increasing trends with an increase in altitude.
However, the recharge proportion of precipitation was higher than those of the



supra-permafrost water and glacier and snow meltwater, and also showed a decreasing
trend from low to high altitude in the final ablation period in 2016. The proportion of
glacier and snow meltwater increased with an increase in altitude, but the recharge
proportion of supra-permafrost water was stable with the change in altitude in the
final ablation period in 2016. The proportion of supra-permafrost water was 50.53%
in the initial and final ablation periods. Meanwhile, supra-permafrost water was the
main recharge source of runoff, followed by precipitation in the total ablation period
from 2016 to 2018, and the proportions of precipitation and supra-permafrost water
were 36.13% and 36.66%, respectively. The recharge proportion of glacier and snow
meltwater was higher in the total ablation period than those in the initial and final
ablation periods, with a proportion of 19.44% in the initial and final ablation periods
and 27.30% in the total ablation period. Compared with the glacier permafrost area
river, the recharge proportion of supra-permafrost water was higher for the permafrost
area river than that for the glacier permafrost area river (42.21%). The recharge
proportion of supra-permafrost water was 69.54%. Just as with the glacier permafrost
area river, the supra-permafrost water was the important recharge source to the runoff
in the initial and final ablation periods, and the proportion was 80.97% in the initial
and final ablation periods. Meanwhile, the proportion of the supra-permafrost water
was 61.92% in the total ablation period. The proportion was higher than that for
precipitation (24.13%) in the same period. In general, the supra-permafrost water was
the major recharge source for the permafrost area river in the study area. Meanwhile,
the glacier and snow meltwater contributed little to the permafrost area river in the





initial and final ablation periods. For the mainstream, the proportion was 35.93% in
initial and final ablation periods, and 45.55% in the total ablation period. However,
the proportion was 47.49% in the initial and final ablation periods, and 36.47% in the
total ablation period. The proportion of glacier and snow meltwater for the
mainstream (16.59%) was higher than that for the permafrost area river (3.25%) but
was lower than that for the glacier permafrost area river (19.44%) in the initial and
final ablation periods. Meanwhile, the proportion of glacier and snow meltwater for
the mainstream (17.98%) was higher than that for the permafrost area river (13.95%)
but was lower than that for the glacier permafrost area river (27.30%) in the total
ablation period.

**Acknowledges**

This study was supported by National "Plan of Ten Thousand People" Youth Top
Talent Project, the Second Tibetan Plateau Scientific Expedition and Research
Program(STEP), Grant No. 2019QZKK0405, the Youth Innovation Promotion
Association, CAS (2013274), Open funding from the Key Laboratory of Mountain
Hazards and Earth Surface Process the open funding from State Key Laboratory of
Loess and Quaternary Geology (SKLLQG1814).

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



**Tables:**

Table 1 The correlation analysis of $\delta^{18}O$ and d-excess and meteorological factors in

the fixed point (TTH and ZMD) from March,16 to July, 18.





Table 1 The correlation analysis of δ¹⁸O and d-excess and meteorological factors in
the fixed point (TTH and ZMD) from March,16 to July, 18.

|  | Precipitation (mm) | Temperature (℃) | Ralative humidity (%) | Evaporation (mm) | δ¹⁸O(‰) | D-excess (‰) |
|---|---|---|---|---|---|---|
| Precipitation(mm) | 1 |  |  |  |  |  |
| Temperature(℃) | 0.853** | 1 |  |  |  |  |
| Ralative humidity(%) | 0.760** | 0.836** | 1 |  |  |  |
| Evaporation(mm) | 0.658** | 0.865** | 0.586** | 1 |  |  |
| δ¹⁸O(‰) | -0.518** | -0.671** | -0.555** | -0.636** | 1 |  |
| D-excess(‰) | 0.500** | 0.602** | 0.524** | 0.533** | -0.568** | 1 |

Note: **, Correlation is significant at the 0.01 level (2-tailed).


**Figures:**


Fig.1 The map of the study area and the sampling point of river water in different
ablation period
(Fig.1a was the detail location of the study area in China and Asian and the distribution of fixed
point for precipitation, river water and glacier and snow meltwater; Fig.1b was the distribution of
sampling point in initial ablation in 2016; Fig.1c was the distribution of sampling point in ablation
in 2016; Fig.1d was the distribution of sampling point in end ablation in 2016; Fig.1e was the
distribution of sampling point in ablation in 2017; Fig.1f was the distribution of sampling point in
ablation in 2018)
Fig.2 Variation of meteorological factors during sampling period
(Shadow represents the ablation period)
Fig.3 Temporal variation of $\delta^{18}O$ and d-excess during the sampling period in study
area
(This figure mainly showed the temporal variation of $\delta^{18}O$ and d-excess for different type runoff
based on different ablation in 2016 and strong ablation from 2016 to 2018; Fig.2a, b, c showed the
change of $\delta^{18}O$ and d-excess in different ablation period for mainstream, glacier and snow runoff
and river in permafrost area; Fig.2d, e, f showed the change of $\delta^{18}O$ and d-excess in ablation
period from 2016 to 2018 for mainstream, glacier and snow runoff and river in permafrost area)
Fig.4 Spatial variation of $\delta^{18}O$ based on different ablation in 2016 and ablation from
2016 to 2018
Fig.5 Spatial variation of d-excess based on different ablation in 2016 and ablation
from 2016 to 2018





Fig.6 The variation of $\delta^{18}O$ and d-excess with the altitude change in study area
(Fig.6a was the variation of $\delta^{18}O$ and d-excess with the altitude change for mainstream; Fig.6b
was the variation of $\delta^{18}O$ and d-excess with the altitude change for river in glacier permafrost
area;Fig.6c was the variation of $\delta^{18}O$ and d-excess with the altitude change for river in permafrost
area; IA in 2016 represents Initial ablation in 2016; A in 2016 represents Ablation in 2016; EA in
2016 represents End ablation in 2016; A in 2017 represents Ablation in 2017; A in 2018
represents Ablation in 2018)
Fig.7 The distribution of $\delta D$ and $\delta^{18}O$ for river water among other water bodies in
study area
(Fig.7a was the plot of $\delta^{18}O$ for river water in different type, supra-permafrost water, glacier snow
meltwater and precipitation; Fig.7b was the plot of $\delta D$ for river water in different type,
supra-permafrost water, glacier snow meltwater and precipitation; Fig.7c was the plot of $\delta D$
versus $\delta^{18}O$ for river water, supra-permafrost water, glacier snow meltwater and precipitation)
Fig.8 Three end element diagram using the mean values of $\delta^{18}O$ and d-excess for river
water in different ablation in 2016 and ablation from 2016 to 2018
Fig.9 Recharge proportion from possible sources to river water in different altitude
during different ablation in 2016 and ablation from 2016 to 2018
Fig.10 The variation of location evaporation line (LEL) of river water based on
different ablation in 2016 and ablation from 2016 to 2018
Fig.11 Conceptual model map of the recharge form and proportion of the river water
in different ablation period
(Dark green represents the basin of river in permafrost area; Gray and light green represents the





basin of the river in glacier permafrost area)









































Fig.1

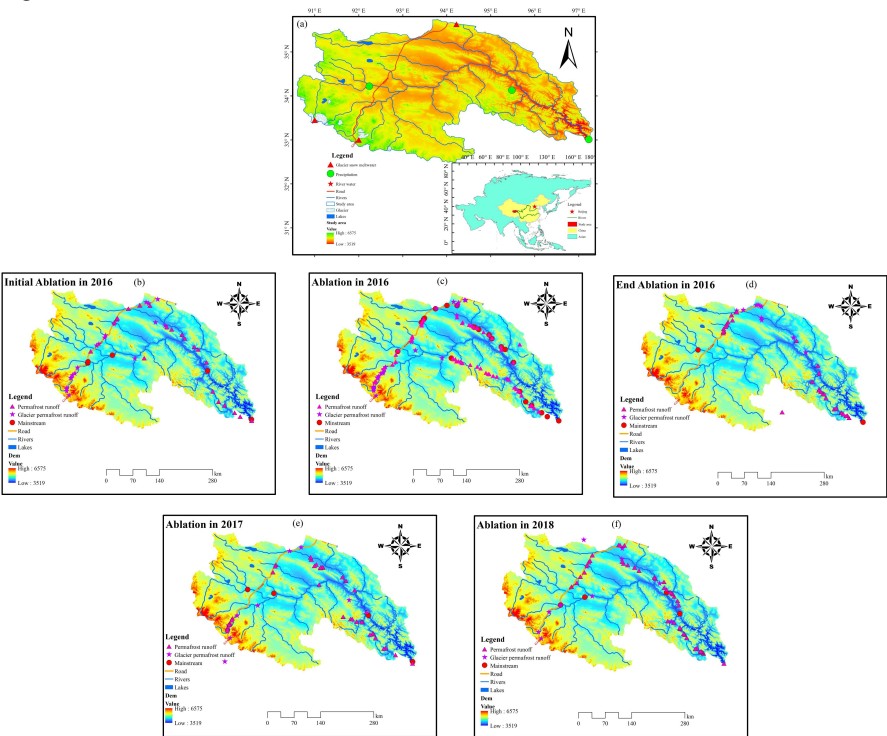


Fig.1 The map of the study area and the sampling point of river water in different
ablation period (Fig.1a was the detail location of the study area in China and Asian and the
distribution of fixed point for precipitation, river water and glacier and snow meltwater; Fig.1b
was the distribution of sampling point in initial ablation in 2016; Fig.1c was the distribution of
sampling point in ablation in 2016; Fig.1d was the distribution of sampling point in end ablation in
2016; Fig.1e was the distribution of sampling point in ablation in 2017; Fig.1f was the distribution
of sampling point in ablation in 2018)





Fig.2

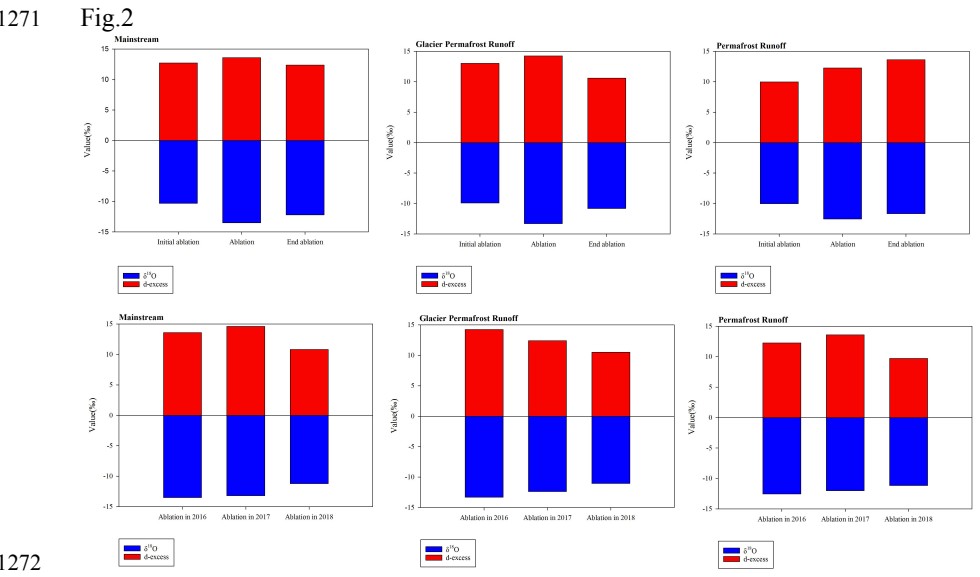


Fig.2 Variation of meteorological factors during sampling period (Shadow represents the

ablation period)




Fig.3

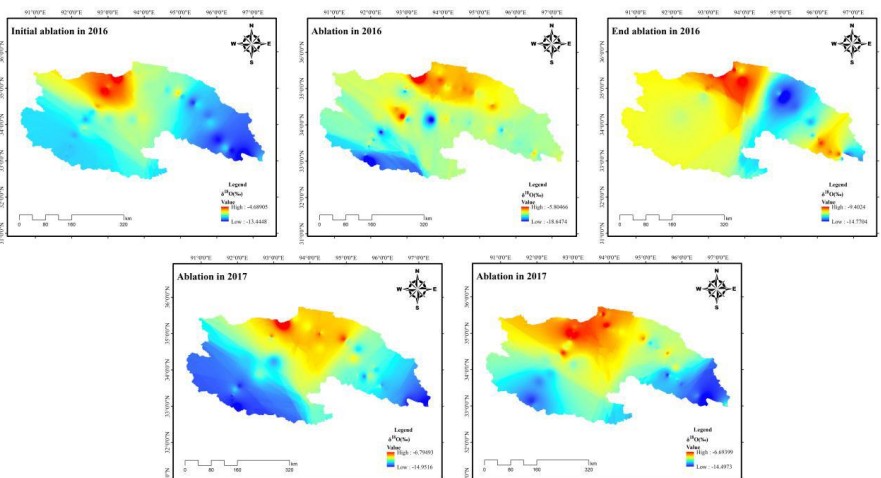


Fig.3 Temporal variation of δ$^{18}$O and d-excess during the sampling period in study
area (This figure mainly showed the temporal variation of δ$^{18}$O and d-excess for different type
runoff based on different ablation in 2016 and strong ablation from 2016 to 2018; Fig.2a, b, c
showed the change of δ$^{18}$O and d-excess in different ablation period for mainstream, glacier and
snow runoff and river in permafrost area; Fig.2d, e, f showed the change of δ$^{18}$O and d-excess in
ablation period from 2016 to 2018 for mainstream, glacier and snow runoff and river in permafrost
area)





Fig.4

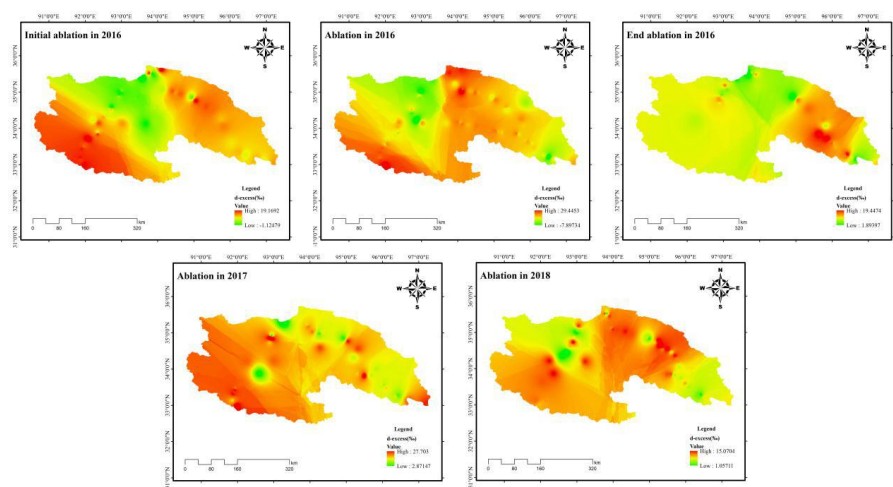


Fig.4 Spatial variation of δ$^{18}$O based on different ablation in 2016 and ablation from

2016 to 2018





























Fig.5

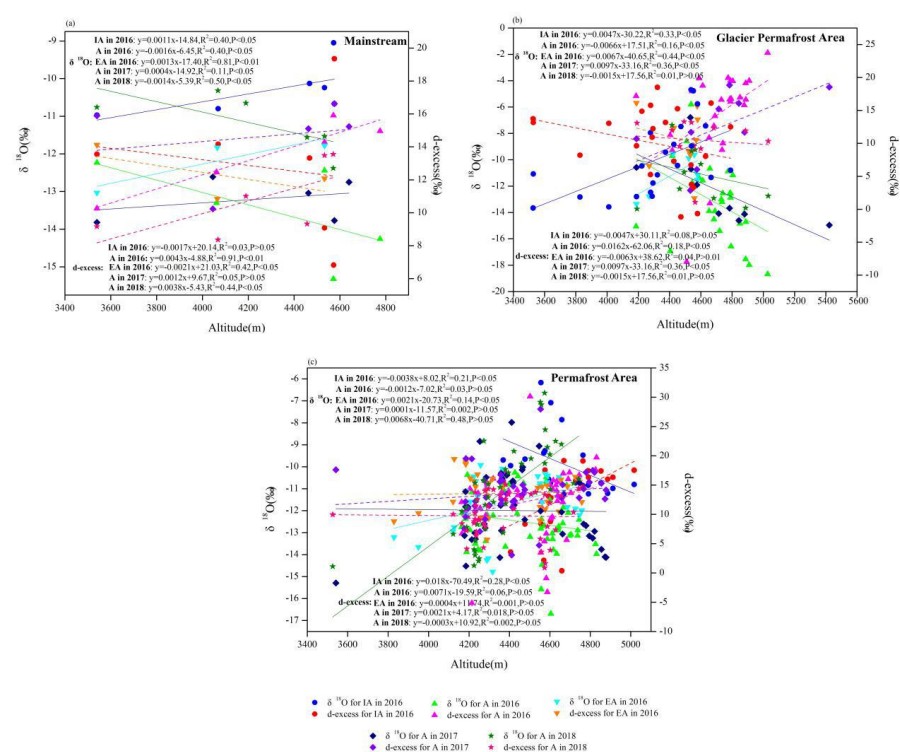


Fig.5 Spatial variation of d-excess based on different ablation in 2016 and ablation

from 2016 to 2018




Fig.6

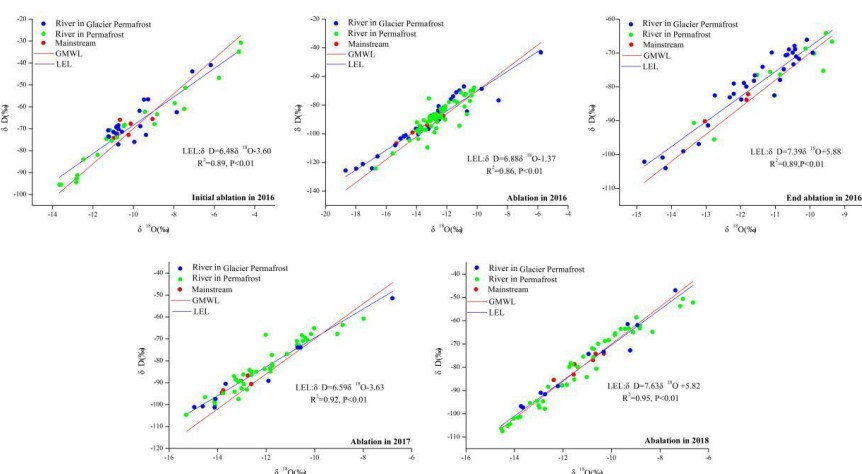


Fig.6 The variation of δ18O and d-excess with the altitude change in study area

(Fig.6a was the variation of δ18O and d-excess with the altitude change for mainstream; Fig.6b
was the variation of δ18O and d-excess with the altitude change for river in glacier permafrost
area;Fig.6c was the variation of δ18O and d-excess with the altitude change for river in permafrost
area; IA in 2016 represents Initial ablation in 2016; A in 2016 represents Ablation in 2016; EA in
2016 represents End ablation in 2016; A in 2017 represents Ablation in 2017; A in 2018

represents Ablation in 2018)


Fig.7

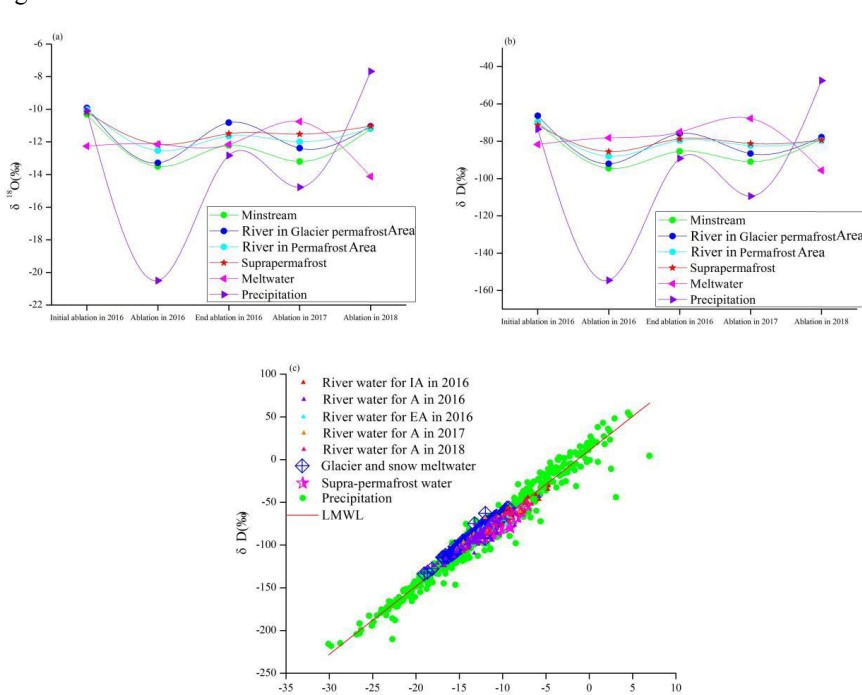



Fig.7 The distribution of δD and δ¹⁸O for river water among other water bodies in
study area (Fig.7a was the plot of δ¹⁸O for river water in different type, supra-permafrost water,
glacier snow meltwater and precipitation; Fig.7b was the plot of δD for river water in different
type, supra-permafrost water, glacier snow meltwater and precipitation; Fig.7c was the plot of δD
versus δ¹⁸O for river water, supra-permafrost water, glacier snow meltwater and precipitation)





Fig.8

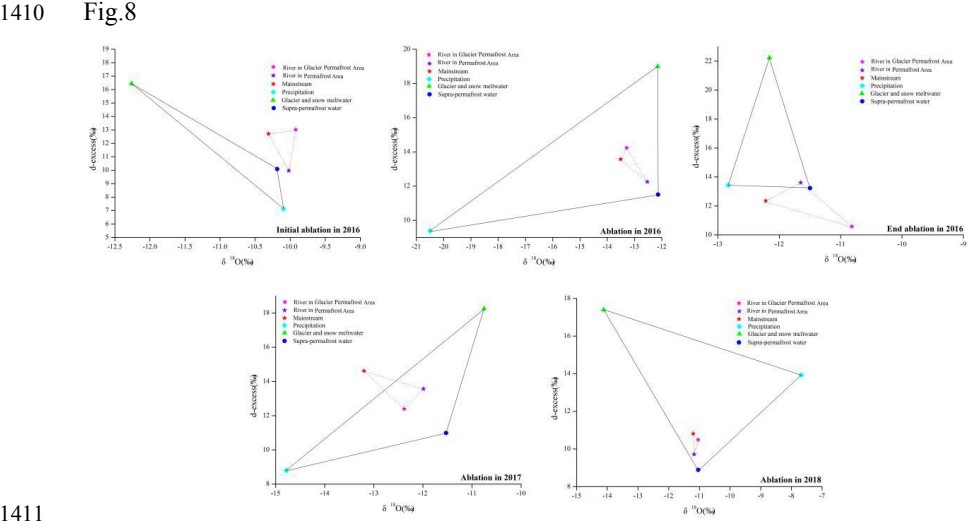


Fig.8 Three end element diagram using the mean values of $\delta^{18}O$ and d-excess for river

water in different ablation in 2016 and ablation from 2016 to 2018





























Fig.9

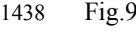

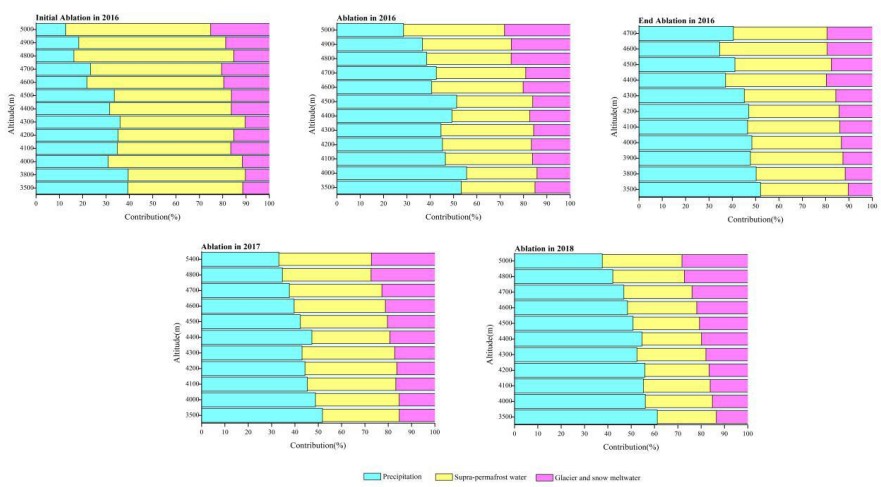


Fig.9 Recharge proportion from possible sources to river water in different altitude

during different ablation in 2016 and ablation from 2016 to 2018






Fig.10

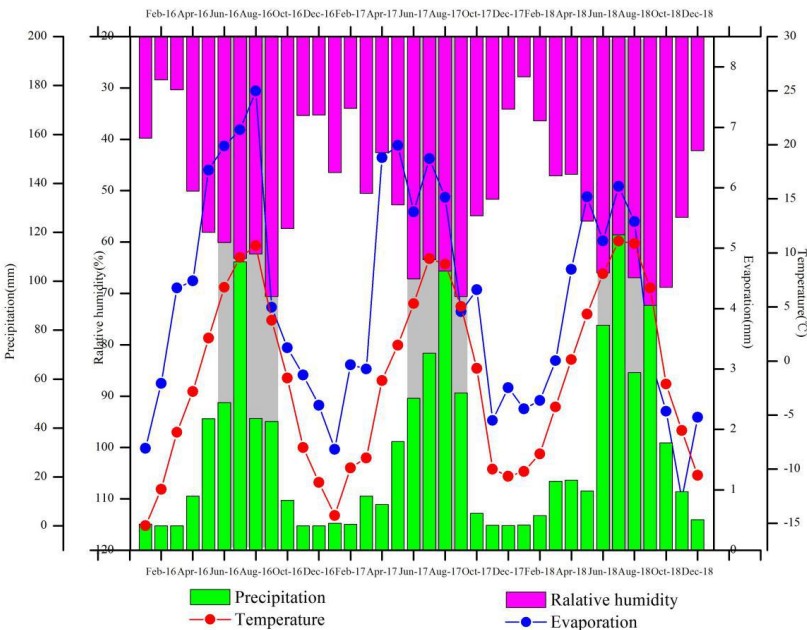


Fig.10 The variation of location evaporation line (LEL) of river water based on
different ablation in 2016 and ablation from 2016 to 2018



Fig.11

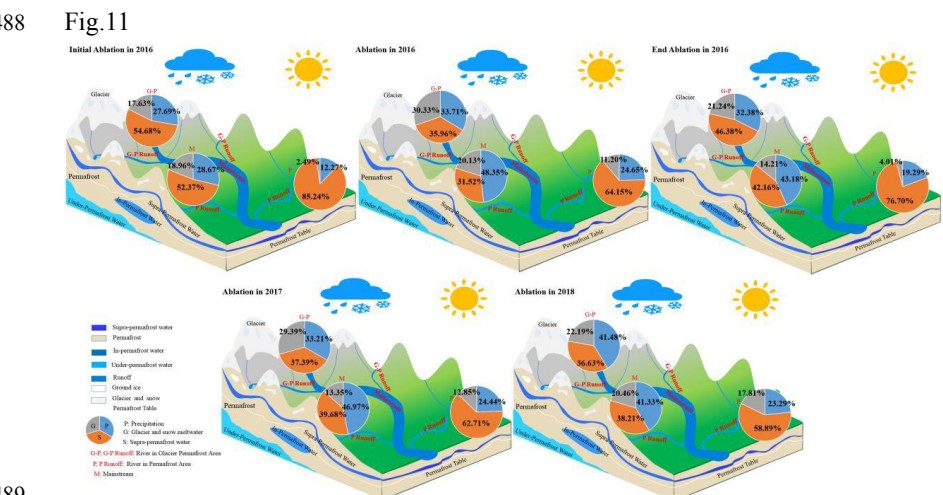


Fig.11 Conceptual model map of the recharge form and proportion of the river water
in different ablation period (Dark green represents the basin of river in permafrost area; Gray

and light green represents the basin of the river in glacier permafrost area)