# Peer review of "Hydrological and Runoff Formation Processes Based on Isotope Tracing During Ablation Period in the Source Regions of Yangtze River"

_Hydrology and Earth System Sciences, 2019_

## Referee Comment (RC1) · Anonymous Referee #1 · 23 Nov 2019

Review of Li et al.,

This manuscript presents a meaningful insight into the hydrological processes in permafrost-affected cold areas. The authors carried out solid experimental works and got valuable isotopic data for understanding the hydrological processes. The influence factors on stable isotopes in water were analyzed and the contributions of each runoff component were quantified, which makes this manuscript worth publishing.

However, there are still some moderate to major issues that need to be improved before published on HESS. The written English needs to be improved thoroughly, and some results were not presented in a clear way. A colleague who has English as first

language would be helpful to read through the manuscript and edit it. Besides, most results can be derived from data directly, but the reason and deeper phenomenon were not addressed enough. Last, there are still some technical issues on the calculation processes. I would outline them in detail below.

Major/Moderate issues: 1) There are a few errors about language, data or figure. Please check the text thoroughly, and I only list some obvious examples here: a. Some figures are not referred correctly in the main text (e.g., L369, Fig. 6 should be Fig. 5), and the title and description of some figures are wrong (e.g., L1371, the title and description of Fig. 6 should be for Fig. 5). b. L641: The proportion of supra-permafrost in the last three periods are same as precipitation, and the total proportion of all the sources in these three periods are not one. c. Fig. 1a: The red star represents river water, but there is no red star in the map. 2) Some concepts are not adequately clarified, making the result confusing. It would be better to clarify the exact time period of the initial/total/final ablation period in 2016 and ablation in 2017 and 2018. It would also be helpful to map the range of mainstream, the glacier permafrost area and the permafrost area. 3) There are paragraphs of words describing the number itself in main text (e.g., L274-302, L462-485), but they provide little useful information and make the result confusing. It would be better to list the numbers in a table and only give important conclusion, implication or explanation in the main text. 4) The "anti-effect" (anti-altitude effect, the negative correlation between isotope and temperature/evaporation) is an interesting phenomenon worth exploring. The discussion about this still lacks detail. How did meteorological factors influence each source directly and indirectly, and finally resulted in the negative correlation? 5) The 4.3 section seems to be a review of hydrological significance of permafrost, rather than a discussion. Please discuss more about the implication of your result, e.g., what kind of significance can you find or confirm according to your data. 6) The conclusion section is too detailed. Please just present the most importation finding here, rather than list so many numbers.

Minor issues: 1) Title: Better to use the source regions of Yangtze river, rather than

Third Polar Region 2) L33: "The effects of altitude" on what? Please clarify. 3) L108: "because it is affected by this", please rewrite. 4) L140-L145: the sentence is too long, please simplify 5) L149-151: How does the result of this article benefit the rational development? Please tell some implications of the results in the discussion section. 6) L161: Please use Salween River and Lantsang River 7) L227: Please use the full name in title 8) L298: "was different than"? 9) L410-L412: Please discuss in more details about the comprehensive influence 10) L437: What do you mean by "river water in different types of water"? 11) L458-L461: The sentence "Owing to . . ." does not has a main clause, please rewrite. 12) L481: I think there should not be a "so" here. 13) L498: I notice that the average contribution here is an arithmetic mean value, which is not reasonable. I think an amount-weighted value will be better. 14) L588-L590: "evaporation process" in L588 and "evaporation loss" in L590 are repeating. 15) L600: I think you are analyzing the stable isotopes in river water. But the river water were not sampled continuously according to the Method section. Please make it clear. 16) L647-L651: same as 13), please use amount-weighted value. 17) L711: What do you mean by "different ablation periods and strong ablation periods"? 18) Figures: Again, please check the name of figures and make sure that they are referred to correctly in main text. 19) Fig7: Please add an error bar to represent the range of isotope in water during each period. 20) Fig11: Is it really necessary to use 5 subfigures? I think they are a same figure apart from the proportion.

---

## Referee Comment (RC2) · Anonymous Referee #2 · 29 Nov 2019

The paper of Li et al. "Hydrological and Runoff Formation Processes Based on Isotope Tracing During Ablation Period in the Third Polar Region" investigates the hydrological and runoff formation processes of river water in the source regions of the Yangtze river during different ablation episodes in 2016 and the ablation period from 2016 to 2018. In particular, the authors discuss the temporal and spatial variations of isotopes in different tributary rivers under the background of climate warming and their influencing factors by using the methods of field observation, experimental testing, stable isotope tracing, and analytical modeling of end-element mixed runoff. In general, I like the idea of understanding the hydrological and runoff formation processes of river water during different ablation period. Also, I think that the data obtained with this study have high

potential interest for the scientific community. Therefore it is worth publishing this article in HESS after revising the following minor revisions. 1. In the abstract precise in the method used. 2. The aim of the study should be exhibited in the introduction section. 3. Line 59-61. "The runoff system in the source area of the Yangtze River consists of alpine glaciers, snow, frozen soil, and liquid precipitation. ". Delete this 4. Line 64-68. "Therefore, studying changes in the composition of runoff and its hydrological effect in cold areas can not only consolidate theories on runoff research, prediction, and adaptation, but also have important practical significance for construction, industry, and agriculture in cold regions" - please rephrase it. 5. Line 109-110. The ground temperature of the permafrost increases, causing it to melt significantly. - rewrite this statement 6. Line 227. EMMA is a section name - required full name not abbreviation. 7. Please change "final ablation" into "end ablation", and change "total ablation" into "ablation". 8. Line 274-276. Sentence is badly written, please rephrase. 9. Line 349. " However, all regions exhibited high ablation, especially in the Tanggula Mountains," please rephrase. 10. The conclusion section is too long, please rewrite. 11. Fig. 8. Small plot inside is unreadable.

---

## Referee Comment (RC3) · Anonymous Referee #3 · 6 Dec 2019

This manuscript presented interesting work on detecting hydrological processes via stable isotope technique conducted in the source area of the Yellow River, where undergoing permafrost degradation caused by climate changes. However, some major issues with the isotope data interpretation, the basis of hydrograph separation and the model calculations, which brought in large uncertainties. Meanwhile, in the discussion section, especially in 4.3, the authors seemed to simply put on existed references or just to repeat reporting similar opinions and reviews from previous studies to support their results, which resulted in the lack of novelty and scientific significances. How the data and results presented in this manuscript can defend for the permafrost hydrology. Besides, there was no discussions on the glacier melting.

[Figure]

Overall, I feel sorry to say that the current quality of this manuscript cannot reach the requirement to be published in HESS, as it did not clearly focus on the "Hydrological and Runoff Formation Processes", nor solve the evolution mechanism of regional runoff involved with climate changes, permafrost degradation, glacier hydrology. I hope the authors can rewrite their manuscript, not only to improve the writing skills and English expressions, but also to significantly contribute to new hydrological insights.

Major concerns: 1. There is no clear $\delta 2H$- $\delta 18O$ space to show the isotopic differences between precipitation, runoff water, permafrost meltwater, glacial meltwater as well as no description on the isotopically comparisons.

2. The EMMA was based on $\delta 18O$ and $\delta d$-excess, however, $\delta d$-excess= $\delta 2H$-8$\delta 18O$, the second tracer was partially relied on the first tracer. According to the basic principles of hydrograph separation (J. Klaus, J.J. McDonnell; Hydrograph Separation Using Stable Isotopes: Review and Evaluation, Journal of Hydrology), using $\delta 18O$ and $\delta d$-excess to do three-sources hydrograph was very weak to achieve reliable results.

3. The authors seemed to use single average isotopic content to represent each source (precipitation, permafrost, glacier). However, to estimate the proportions of each component in areas influenced by different permafrost/glacier degradations without considering the spatial and temporal heterogeneity of isotopes as well as evaporation effects along the water flow (changing isotope values) in such extensive watershed might cause great uncertainties.

4. The uncertainties should be addressed. Many factors instead of the only measurement error.

Minor comments: Too many grammatical and word errors, as well as mistakes in graphs and captions. Authors should check their manuscript very carefully and ask for some native speaker to edit to make paper readable before submission.

530, 2019.

---

## Author Comment (AC1) · 8 Jan 2020

Response to Reviewer#2 Anonymous Referee #2 The paper of Li et al. "Hydrological and Runoff Formation Processes Based on Isotope Tracing During Ablation Period in the Third Polar Region" investigates the hydrological and runoff formation processes of river water in the source regions of the Yangtze river during different ablation episodes in 2016 and the ablation period from 2016 to 2018. In particular, the authors discuss the temporal and spatial variations of isotopes in different tributary rivers under the background of climate warming and their influencing factors by using the methods of field observation, experimental testing, sta-

ble isotope tracing, and analytical modeling of end-element mixed runoff. In general, I like the idea of understanding the hydrological and runoff formation processes of river water during different ablation period. Also, I think that the data obtained with this study have highpotential interest for the scientific community. Therefore it is worth publishing this article in HESS after revising the following minor revisions. Thank you very much for your comments. 1.In the abstract precise in the method used. Thank you very much for your comments. The abstract section has been revised as: "This study focused on the hydrological and runoff formation processes of river water by using stable isotope tracing in the source regions of the Yangtze river during different ablation episodes in 2016 and the ablation period from 2016 to 2018. The effects of altitude on stable isotope characteristics for the river in the glacier permafrost area were greater than for the mainstream and the permafrost area during the total ablation period in 2016. There was a significant negative correlation (at the 0.01 level) between precipitation and $\delta$18O, while a significant positive correlation was evident between precipitation and d-excess. More interestingly, significant negative correlations appeared between $\delta$18O and temperature, relative humidity, and evaporation. A mixed segmentation model for end-members was used to determine the proportion of the contributions of different water sources to the target water body. The proportions of precipitation, supra-permafrost water, and glacier and snow meltwater for the mainstream were 41.70%, 40.88%, and 17.42%, respectively. The proportions of precipitation, supra-permafrost water, and glacier and snow meltwater were 33.63%, 42.21%, and 24.16% for the river in the glacier permafrost area and 20.79%, 69.54%, and 9.67%, respectively, for that in the permafrost area. The supra-permafrost water was relatively stable during the different ablation periods, becoming the main source of runoff in the alpine region, except for precipitation, during the total ablation period." 2.The aim of the study should be exhibited in the introduction section. Thank you very much for your comments. The aim of the study has been added as: "Based on the conversion signals of stable isotopes in each link of the runoff process, at first, this study further explores the hydraulic relations, recharge-drainage relations and their transformation

paths, and the processes of each water body. Furthermore, this study determines the composition of runoff, quantifies the contribution of each runoff component to different types of tributaries. Finally, this study analyzes the hydrological effects of the temporal and spatial variation of runoff components. " 3.Line 59-61. "The runoff system in the source area of the Yangtze River consists of alpine glaciers, snow, frozen soil, and liquid precipitation. ". Delete this Thank you very much for your comments. I have deleted it. 4.Line 64-68. "Therefore, studying changes in the composition of runoff and its hydrological effect in cold areas can not only consolidate theories on runoff research, prediction, and adaptation, but also have important practical significance for construction, industry, and agriculture in cold regions" - please rephrase it. Thank you very much for your comments. It has been revised as: "Therefore, the study on the composition change of runoff and its hydrological effect in cold areas can not only consolidate theories on runoff research, prediction, and adaptation, but also have important practical significance for construction, industry, and agriculture in cold regions " 5.Line 109-110. The ground temperature of the permafrost increases, causing it to melt significantly. - rewrite this statement Thank you very much for your comments. It has been revised as: " 6.Line 227. EMMA is a section name - required full name not abbreviation. Thank you very much for your comments. It has been revised as: " 2.3 End-Member Mixing Analysis" 7.Please change "final ablation" into "end ablation", and change "total ablation" into "ablation". Thank you very much for your comments. I have changed it. 8.Line 274-276. Sentence is badly written, please rephrase. Thank you very much for your comments. It has been revised as: "As shown in Fig. 2, Stable isotope characteristics of $\delta$18O and d-excess was different during different ablation for the different types of runoff. " 9.Line 349. " However, all regions exhibited high ablation, especially in the Tanggula Mountains," please rephrase. Thank you very much for your comments. It has been revised as: "However, all regions except for areas in the eastern region where the ablation was low during the ablation period in 2017 exhibited high ablation especially Tanggula Mountains." 10.The conclusion section is too long, please rewrite. Thank you very much for your comments. The conclusion section has

been revised as: "Through systematically analysis of the characteristics of $\delta$18O, $\delta$D, and d-excess of river water in the different ablation periods in 2016 and the ablation periods from 2016 to 2018, the results were as follows. The temporal and spatial characteristics of stable isotopes of river water were significant in the study area. The $\delta$18O in mainstream was more negative than that in the glacier permafrost area river and permafrost area river. The influence of evaporation on isotope and d-excess is only prevalent in some places, such as the central and northern parts of the study area in the initial ablation and total ablation periods. However, the influence of evaporation on isotope and d-excess is prevalent in most places except the southeastern part of the study area. Meanwhile, this results also indicated that there may be a hysteresis for the influence of meteorological factors on isotopes and d-excess. The altitude effect is only present during the ablation periods in 2016 and 2018, and the altitude effect was $-0.16$‰100Ăm (pĂ<Ă0.05) and $-0.14$‰100Ăm (pĂ<Ă0.05). The slope of LEL for river water showed an increasing trend from initial ablation to final ablation in 2016. Meanwhile, the intercept of LEL for river water also increased from the initial ablation to the final ablation period. Moreover, the mixed segmentation model of the end-member is used to determine the contribution proportion of different water sources to the target water. The results showed that the supra-permafrost water was the major recharge source for the permafrost area river in the study area. Meanwhile, the glacier and snow meltwater contributed little to the permafrost area river in the initial and final ablation periods. For the mainstream, the proportion was 35.93% in initial and final ablation periods, and 45.55% in the total ablation period. However, the proportion was 47.49% in the initial and final ablation periods, and 36.47% in the total ablation period. The proportion of glacier and snow meltwater for the mainstream (16.59%) was higher than that for the permafrost area river (3.25%) but was lower than that for the glacier permafrost area river (19.44%) in the initial and final ablation periods. Meanwhile, the proportion of glacier and snow meltwater for the mainstream (17.98%) was higher than that for the permafrost area river (13.95%) but was lower than that for the glacier permafrost area river (27.30%) in the total ablation period. ".

11.Fig. 8.Small plot inside is unreadable. Thank you very much for your comments. It has been revised as:

Please also note the supplement to this comment:
https://www.hydrol-earth-syst-sci-discuss.net/hess-2019-530/hess-2019-530-AC1-supplement.pdf

---

## Author Comment (AC2) · 8 Jan 2020

Response to Reviewer#1 Interactive comment on "Hydrological and Runoff Formation Processes Based on Isotope Tracing During Ablation Period in the Third Polar Region"by Zong-Jie Li et al. Anonymous Referee #1

Review of Li et al., This manuscript presents a meaningful insight into the hydrological processes in permafrost-affected cold areas. The authors carried out solid experimental works and got valuable isotopic data for understanding the hydrological processes. The influence factors on stable isotopes in water were analyzed and the contributions

of each runoff component were quantified, which makes this manuscript worth publishing. However, there are still some moderate to major issues that need to be improved before published on HESS. The written English needs to be improved thoroughly, and some results were not presented in a clear way. A colleague who has English as first language would be helpful to read through the manuscript and edit it. Besides, most results can be derived from data directly, but the reason and deeper phenomenon were not addressed enough. Last, there are still some technical issues on the calculation processes. I would outline them in detail below. Thank you very much for your comments. Major/Moderate issues: 1)There are a few errors about language, data or figure. Please check the text thoroughly, and I only list some obvious examples here: Thank you very much for your comments. I have checked the text thoroughly. a.Some figures are not referred correctly in the main text (e.g., L369, Fig. 6 should be Fig.5), and the title and description of some figures are wrong (e.g., L1371, the title and description of Fig. 6 should be for Fig. 5). Thank you very much for your comments. "Fig.6" has been "Fig.5". The title and description of all figures has been revised. b.L641: The proportion of supra-permafrost in the last three periods are same as precipitation, and the total proportion of all the sources in these three periods are not one. Thank you very much for your comments. It has been revised as: "the proportion of supra-permafrost water in the initial, total, and final ablations in 2016, the total ablation in 2017, and the total ablation in 2018 were 54.68%, 35.96%, 46.38%, 37.39%, and 36.63%, respectively." c. Fig. 1a: The red star represents river water, but there is no red star in the map. Thank you very much for your comments. Fig.1 has been revised as:

2) Some concepts are not adequately clarified, making the result confusing. It would be better to clarify the exact time period of the initial/total/final ablation period in 2016 and ablation in 2017 and 2018. It would also be helpful to map the range of mainstream, the glacier permafrost area and the permafrost area. Thank you very much for your comments. I have added the concepts for the exact time period of the initial/total/final ablation period in the section of 2.2 Sample Collection, as follows: "In this study, the

initial ablation period is from May to June, the strong ablation period is from July to August, and the end ablation period is from September to October. ". 3)There are paragraphs of words describing the number itself in main text (e.g., L274-302, L462-485), but they provide little useful information and make the result confusing. It would be better to list the numbers in a table and only give important conclusion, implication or explanation in the main text. Thank you very much for your comments. L274-302 has been revised as: "For the mainstream, the $\delta$18O in initial ablation was higher than end ablation, while the ablation period was the lowest. But $\delta$18O in ablation period showed decreasing trend from 2016 to 2018. With the same as $\delta$18O, d-excess in the different ablation periods was different (Fig. 2a, d). For the river in the glacier permafrost area, the order of $\delta$18O for the different ablation periods and the ablation period from 2016 to 2018 was the same as the mainstream order, but the values of $\delta$18O were different for the mainstream (Fig. 2b, e). For the river in the permafrost area, the variation $\delta$18O for the different ablation periods and ablation from 2016 to 2018 was the same as for the mainstream and the river in the glacier permafrost area. However, the order of d-excess was different for the river in the permafrost area and the glacier permafrost area (Fig. 2c, f).". L462-485 has been revised as: "As shown in Fig. 8, according to the locations of the different types of water and the distance from other water bodies, which reflected the mixed recharge of three water bodies, supra-permafrost water was the first end element, precipitation was the second end element, and glacier and snow meltwater was the third end element. However, the different runoffs likely have different recharge sources and different recharge proportions. The glacier permafrost area river comprised glacier and snow meltwater more in the total ablation period than in other periods. Compared with the permafrost area river and the glacier permafrost area river, the mainstream was governed by the supra-permafrost water in the initial ablation period while containing nearly equal proportions of supra-permafrost water and precipitation in the final ablation period. ". 4)The "anti-effect" (anti-altitude effect, the negative correlation between isotope and temperature/evaporation) is an interesting phenomenon worth exploring. The discussion about this still lacks detail. How did

meteorological factors influence each source directly and indirectly, and finally resulted in the negative correlation? Thank you very much for your comments. It has been added asïijŻ"For the phenomenon of anti-altitude effect, the following reasons can explain this phenomenon: on the one hand, in the source area of the river, the stable isotope concentration of precipitation and glacier snow meltwater is relatively low and the value of groundwater in the permafrost active layer is relatively positive due to the influence of soil evaporation; On the other hand, the more the inflow of precipitation, the greater the contribution of precipitation. So there is an obvious diluting effect of biotin, which makes the concentration more negative." 5)The 4.3 section seems to be a review of hydrological significance of permafrost, rather than a discussion. Please discuss more about the implication of your result, e.g., what kind of significance can you find or confirm according to your data. Thank you very much for your comments. The 4.3 section has been revised as: "The source region of the Yangtze River is a typical permafrost area. The permafrost area is 107619.13 km2, which accounting for 77% of the total area. The seasonal frozen soil is mainly distributed in the valley area, with an area of 30754.34 km2. Field observation and research confirmed that most of the precipitation in permafrost area is frozen on the ground or used to recharge the deficit of soil water, and does not directly form runoff in permafrost area. Under the background of permafrost degradation, the area of permafrost is gradually shrinking and the thickness of permafrost is gradually decreasing with the increase of the thickness of active layer. The degradation of ice rich permafrost in the cold regions has an important contribution to the development of surface runoff and hot karst lakes. Due to the decrease of permafrost water storage capacity in the Qinghai Tibet Plateau, the availability of water resources will be reduced in dry season, and the increase of water melting may lead to the increase of flood risk, and the resilience of ecosystem will be reduced through the seasonal changes of river flow and groundwater abundance. All these changes will affect the water resources balance and sustainable development of the Qinghai Tibet Plateau, including the headwaters of major rivers in Asia, including the Yellow River, the Yangtze River, the Salween River, the Mekong River, the Brahmaputra River, the Ganges River, the Indus River, the Ili River, the Tarim River, the Erqis River and the Yenisei River Rivers, these rivers provide fresh water resources for the survival of about 2 billion people.

In brief, the freeze-thaw of soil in the active layer plays an important role in controlling river runoff. The increase in melting depth leads to a decrease in the direct runoff rate and slow dewatering process. The two processes of runoff retreat are the result of soil freeze-thaw in the active layer. Permafrost has two hydrological functions: on the one hand, permafrost is an impervious layer, and it has the function of preventing surface water or liquid water from infiltrating into deep soil; on the other hand, it forms a soil temperature gradient, which makes the soil moisture close to the ice cover. Therefore, changes in the soil water capacity, soil water permeability, and soil water conductivity, as well as the redistribution of water in the soil profile, are caused by the freeze-thaw of the active layer. The seasonal freeze-thaw process of the active layer directly leads to seasonal flow changes in surface water and groundwater, which affects surface runoff. Climate warming is the main driving force in the degradation of cold ecosystems (Wang et al., 2009; Wu et al., 2015; Li et al., 2018; Wang et al., 2019). More importantly, under the background of intense melting, the melting water of the cryosphere has had a significant impact on the hydrological process in the cold region. The hydrological function of groundwater in the permafrost active layer should be paid more attention, especially in the cold region where glaciers are about to subside, its hydrological function needs to be re recognized. The stable isotope characteristics of the cryosphere are more complex than other regions, and its mechanism is more complex Further research is needed."

6)The conclusion section is too detailed. Please just present the most importation finding here, rather than list so many numbers. Thank you very much for your comments. The conclusion section has been revised as: "Through systematically analysis of the characteristics of $\delta$18O, $\delta$D, and d-excess of river water in the different ablation periods in 2016 and the ablation periods from 2016 to 2018, the results were as follows. The

temporal and spatial characteristics of stable isotopes of river water were significant in the study area. The $\delta$18O in mainstream was more negative than that in the glacier permafrost area river and permafrost area river. The influence of evaporation on isotope and d-excess is only prevalent in some places, such as the central and northern parts of the study area in the initial ablation and total ablation periods. However, the influence of evaporation on isotope and d-excess is prevalent in most places except the southeastern part of the study area. Meanwhile, this results also indicated that there may be a hysteresis for the influence of meteorological factors on isotopes and d-excess. The altitude effect is only present during the ablation periods in 2016 and 2018, and the altitude effect was $-0.16$‰100Ăm (pĂ<Ă0.05) and $-0.14$‰100Ăm (pĂ<Ă0.05). The slope of LEL for river water showed an increasing trend from initial ablation to final ablation in 2016. Meanwhile, the intercept of LEL for river water also increased from the initial ablation to the final ablation period. Moreover, the mixed segmentation model of the end-member is used to determine the contribution proportion of different water sources to the target water. The results showed that the supra-permafrost water was the major recharge source for the permafrost area river in the study area. Meanwhile, the glacier and snow meltwater contributed little to the permafrost area river in the initial and final ablation periods. For the mainstream, the proportion was 35.93% in initial and final ablation periods, and 45.55% in the total ablation period. However, the proportion was 47.49% in the initial and final ablation periods, and 36.47% in the total ablation period. The proportion of glacier and snow meltwater for the mainstream (16.59%) was higher than that for the permafrost area river (3.25%) but was lower than that for the glacier permafrost area river (19.44%) in the initial and final ablation periods. Meanwhile, the proportion of glacier and snow meltwater for the mainstream (17.98%) was higher than that for the permafrost area river (13.95%) but was lower than that for the glacier permafrost area river (27.30%) in the total ablation period. ”.

Minor issues: 1)Title: Better to use the source regions of Yangtze river, rather than Third Polar Region Thank you very much for your comments. Title has been revised as: “Hydrological and Runoff Formation Processes Based on Isotope Tracing During

[Figure]

Ablation Period in the Source Regions of Yangtze River" 2)L33: "The effects of altitude" on what? Please clarify. Thank you very much for your comments. It has been revised as "The effects of altitude on stable isotope characteristics for the river in the glacier permafrost area were greater than for the mainstream and the permafrost area during the total ablation period in 2016." 3)L108:"because it is affected by this", please rewrite. Thank you very much for your comments. It has been rewritten as "The regional climate shows a significant warm and wet trend against the background of global climate change. So regional evapotranspiration increases and ice and snow resources exhibit an accelerating melting trend " 4)L140-L145: the sentence is too long, please simplify Thank you very much for your comments. It has been revised as "Based on the conversion signals of stable isotopes in each link of the runoff process, at first, this study further explores the hydraulic relations, recharge-drainage relations and their transformation paths, and the processes of each water body. Furthermore, this study determines the composition of runoff, quantifies the contribution of each runoff component to different types of tributaries. Finally, this study analyzes the hydrological effects of the temporal and spatial variation of runoff components. " 5)L149-151: How does the result of this article benefit the rational development? Please tell some implications of the results in the discussion section. Thank you very much for your comments. I have added as: "The source region of the Yangtze River is a typical permafrost area. The permafrost area is 107619.13 km2, which accounting for 77% of the total area. The seasonal frozen soil is mainly distributed in the valley area, with an area of 30754.34 km2. Field observation and research confirmed that most of the precipitation in permafrost area is frozen on the ground or used to recharge the deficit of soil water, and does not directly form runoff in permafrost area. Under the background of permafrost degradation, the area of permafrost is gradually shrinking and the thickness of permafrost is gradually decreasing with the increase of the thickness of active layer. The degradation of ice rich permafrost in the cold regions has an important contribution to the development of surface runoff and hot karst lakes. Due to the decrease of permafrost water storage capacity in the Qinghai Tibet Plateau, the availability of water

resources will be reduced in dry season, and the increase of water melting may lead to the increase of flood risk, and the resilience of ecosystem will be reduced through the seasonal changes of river flow and groundwater abundance. All these changes will affect the water resources balance and sustainable development of the Qinghai Tibet Plateau, including the headwaters of major rivers in Asia, including the Yellow River, the Yangtze River, the Salween River, the Mekong River, the Brahmaputra River, the Ganges River, the Indus River, the Ili River, the Tarim River, the Erqis River and the Yenisei River Rivers, these rivers provide fresh water resources for the survival of about 2 billion people.

In brief, the freeze-thaw of soil in the active layer plays an important role in controlling river runoff. The increase in melting depth leads to a decrease in the direct runoff rate and slow dewatering process. The two processes of runoff retreat are the result of soil freeze-thaw in the active layer. Permafrost has two hydrological functions: on the one hand, permafrost is an impervious layer, and it has the function of preventing surface water or liquid water from infiltrating into deep soil; on the other hand, it forms a soil temperature gradient, which makes the soil moisture close to the ice cover. Therefore, changes in the soil water capacity, soil water permeability, and soil water conductivity, as well as the redistribution of water in the soil profile, are caused by the freeze-thaw of the active layer. The seasonal freeze-thaw process of the active layer directly leads to seasonal flow changes in surface water and groundwater, which affects surface runoff. Climate warming is the main driving force in the degradation of cold ecosystems (Wang et al., 2009; Wu et al., 2015; Li et al., 2018; Wang et al., 2019). More importantly, under the background of intense melting, the melting water of the cryosphere has had a significant impact on the hydrological process in the cold region. The hydrological function of groundwater in the permafrost active layer should be paid more attention, especially in the cold region where glaciers are about to subside, its hydrological function needs to be re recognized. The stable isotope characteristics of the cryosphere are more complex than other regions, and its mechanism is more complex Further research is needed." 6)L161: Please use Salween River and Lantsang River Thank you very much

for your comments. I have revised. 7)L227: Please use the full name in title Thank you very much for your comments. It has been revised as: " 2.3 End-Member Mixing Analysis" 8) L298: "was different than"? Thank you very much for your comments. It has been revised as: "the order of d-excess was different for the river in the permafrost area and the glacier permafrost area" 9)L410-L412: Please discuss in more details about the comprehensive influence Thank you very much for your comments. It has been revised as: "These results may be due to the comprehensive influence of possible recharge sources and different recharge proportions caused by the influence of meteorological factors. This kind of comprehensive influence is mainly due to the significant seasonality of climate factors in the cold regions, which directly determines the types and contribution proportion of possible recharge sources. Therefore, this result can not be said to be caused by any one factor, but can only be explained by the comprehensive influence of possible recharge sources and different recharge proportions caused by the influence of meteorological factors." 10)L437: What do you mean by "river water in different types of water"? Thank you very much for your comments. It has been revised as: "The distribution of $\delta D$ and $\delta 18O$ for river water among other water bodies are shown in Fig. 8 during the different ablation periods in 2016 and ablation from 2016 to 2018. " 11)L458-L461: The sentence "Owing to : : :" does not has a main clause, please rewrite. Thank you very much for your comments. It has been revised as: "Owing to the two stable isotope concentrations in different water bodies have significant spatial and temporal differences, it can effectively distinguish different water bodies and their mixing relationships. " 12)L481: I think there should not be a "so" here. Thank you very much for your comments. I have deleted "so". 13)L498: I notice that the average contribution here is an arithmetic mean value, which is not reasonable. I think an amount-weighted value will be better. Thank you very much for your comments. This is a good comments. For the accuracy of the results, the stable isotope values of each water body were amount-weighted before calculating the contribution proportion. So we will not be amount-weighted again. 14)L588-L590: "evaporation process" in L588 and "evaporation loss" in L590 are repeating. Thank you very much for your comments. It has been revised as: "The stable isotopes of hydrogen and oxygen in river water are comprehensively affected by the evaporation process, runoff change, precipitation recharge, glacier and snow meltwater recharge and supra-permafrost water in cold regions. " 15)L600:I think you are analyzing the stable isotopes in river water. But the river water were not sampled continuously according to the Method section. Please make it clear. Thank you very much for your comments. I have added the continuous observations at two fixed-point stations in Method section as: " In order to analyze the influence of meteorological factors on the stable isotope in river water, samples were collected once per week at the ZMD and TTH stations throughout the sampling period. A total of 201 river water samples were collected in this study " 16)L647-L651: same as 13), please use amount-weighted value. Thank you very much for your comments. This is a good comments. For the accuracy of the results, the stable isotope values of each water body were amount-weighted before calculating the contribution proportion. So we will not be amount-weighted again. 17)L711: What do you mean by "different ablation periods and strong ablation periods"? Thank you very much for your comments. It has been revised as: "Because of the inherent seasonal variation in precipitation, there were significant changes in precipitation during the different ablation periods. " 18)Figures: Again, please check the name of figures and make sure that they are referred to correctly in main text. Thank you very much for your comments.The name of all figures has been checked. 19)Fig7: Please add an error bar to represent the range of isotope in water during each period. Thank you very much for your comments. It has been revised as:

20) Fig11: Is it really necessary to use 5 subfigures? I think they are a same figure apart from the proportion. Thank you very much for your comments. We really need 5 subfigures here. Because of each subfigures represent different ablation periods. Although each base figure is the same, its recharge proportion is different in different ablation period.

Please also note the supplement to this comment:

[Figure]

https://www.hydrol-earth-syst-sci-discuss.net/hess-2019-530/hess-2019-530-AC2-supplement.pdf

---

## Author Comment (AC3) · 8 Jan 2020

Response to Reviewer#3 Anonymous Referee #3 This manuscript presented interesting work on detecting hydrological processes via stable isotope technique conducted in the source area of the Yellow River, where undergoing permafrost degradation caused by climate changes. However, some major issues with the isotope data interpretation, the basis of hydrograph separation and the model calculations, which brought in large uncertainties. Meanwhile, in the discussion section, especially in 4.3, the authors seemed to simply put on existed references or just to repeat reporting similar opinions and reviews from previous studies

to support their results, which resulted in the lack of novelty and scientific significances. How the data and results presented in this manuscript can defend for the permafrost hydrology. Besides, there was no discussions on the glacier melting. Overall, I feel sorry to say that the current quality of this manuscript cannot reach the requirement to be published in HESS, as it did not clearly focus on the "Hydrological and Runoff Formation Processes", nor solve the evolution mechanism of regional runoff involved with climate changes, permafrost degradation, glacier hydrology. I hope the authors can rewrite their manuscript, not only to improve the writing skills and English expressions, but also to significantly contribute to new hydrological insights. Thank you very much for your comments. Major concerns: 1.There is no clear 2H- 18O space to show the isotopic differences between precipitation, runoff water, permafrost meltwater, glacial meltwater as well as no description on the isotopically comparisons. Thank you very much for your comments. the isotopic differences between river water, supra-permafrost water, glacier snow meltwater, and precipitation analyzed in Section 3.4. Fig. 7 also showed the isotopic differences between river water, supra-permafrost water, glacier snow meltwater, and precipitation. The results showed as: "The distribution of $\delta D$ and $\delta 18O$ for river water among other water bodies are shown in Fig. 7 during the different ablation periods in 2016 and ablation from 2016 to 2018. The results of the distribution of $\delta D$ and $\delta 18O$ of river water indicate the possible recharge sources of river water. However, the $\delta D$ and $\delta 18O$ of river water, supra-permafrost water, glacier snow meltwater, and precipitation exhibited little change during the initial ablation in 2016 (Fig. 7a, b). This phenomenon suggests that precipitation may be the major recharge sources for river water during the initial ablation. A plot of $\delta D$ versus $\delta 18O$ for river and supra-permafrost water, glacier snow meltwater, and precipitation is shown in Fig. 7c. The $\delta D$ and $\delta 18O$ values of glacier and snow meltwater from above the LMWL are the most negative compared to other water bodies. The stable isotope of supra-permafrost water was relatively more positive, located below the LMWL, confirming the influence of strong evaporation. The stable isotope of river water was close to the LMWL, and its concentration value was between precipitation, glacier and snow

meltwater, and supra-permafrost water, reflecting that river water was recharged and affected by multi-source water in the study area. Moreover, the distribution of river water, glacier and snow meltwater, and supra-permafrost water also indicated that there was a hydraulic relationship between the source and target in the different ablation periods in 2016 and ablation from 2016 to 2018."

Fig.7 The distribution of $\delta$D and $\delta$18O for river water among other water bodies in study area 2.The EMMA was based on 18O and d-excess, however, d-excess= 2H-818O, the second tracer was partially relied on the first tracer. According to the basic principles of hydrograph separation (J. Klaus, J.J. McDonnell; Hydrograph Separation Using Stable Isotopes: Review and Evaluation, Journal of Hydrology), using 18O and dexcess to do three-sources hydrograph was very weak to achieve reliable results. Thank you very much for your comments. Hydrograph separation is a widely applied technique that uses the stable isotopes of water (2H and 18O) or other tracers to quantify the contribution of different water sources to streamflow. For its successful application it is critical to adequately characterize these sources (end-members). Although using 18O and d-excess to do three-sources hydrograph was very weak to achieve reliable results, the uncertainty of hydrograph separation results is analyzed systematically in this study. Meanwhile, there are many studies have used 18O and d-excess as tracers to segment runoff (such as: Liu et al., 2008 (Journal of Hydrology); Kong and Pang, 2012(Journal of Hydrology); Penna, D., & van Meerveld, H. I, 2019 (Wiley Interdisciplinary Reviews: Water). 3.The authors seemed to use single average isotopic content to represent each source (precipitation, permafrost, glacier). However, to estimate the proportions of each component in areas influenced by different permafrost/glacier degradations without considering the spatial and temporal heterogeneity of isotopes as well as evaporation effects along the water flow (changing isotope values) in such extensive watershed might cause great uncertainties. Thank you very much for your comments. I have added uncertainty method as: "Uncertainty in hydrograph separation The uncertainty of tracer‐based hydrograph separations can be calculated using the error propagation technique (Genereux, 1998; Klaus & McDonnell, 2013).

This approach considers errors of all separation equation variables. Assuming that the contribution of a specific streamflow component to streamflow is a function of several variables c1, c2, . . ., cn and the uncertainty in each variable is independent of the uncertainty in the others, the uncertainty in the target variable (e.g.,the contribution of a specific streamflow component) is estimatedusing the following equation (Genereux, 1998; Uhlenbrook & Hoeg,2003): (3) where W represents the uncertainty in the variable specified in the ubscript. fx is the contribution of a specific streamflow component x to streamflow. The software package MATLAB is used to apply equation 3 to the different hydrograph separations in this study." And added uncertainty analysis: "Using the approach shown in Equation (3), the uncertainty originating from the variation in the tracers of components and measurement methods could be calculated separately (Uhlenbrook & Hoeg, 2003; Pu et al., 2013). According to the calculations made using Equation (3), the uncertainty was estimated to be 0.07 for the three‐component mixing model in the study region. The uncertainty terms for supra-permafrost water accounted for more than 50.0% of the total uncertainty, indicating that the $\delta$18O and $\delta$D variations of supra-permafrost water accounted for the majority of the uncertainty. Although there is some uncertainty for hydrograph separation, isotope-based hydrograph separations are still valuable tools for evaluating the contribution of meltwater to water resources, and they are particularly helpful for improving our understanding of hydrological processes in cold regions, where there is a lack of observational data. " 4.The uncertainties should be addressed. Many factors instead of the only measurement error. Thank you very much for your comments. I have added as: "Using the approach shown in Equation (3), the uncertainty originating from the variation in the tracers of components and measurement methods could be calculated separately (Uhlenbrook & Hoeg, 2003; Pu et al., 2013). According to the calculations made using Equation (3), the uncertainty was estimated to be 0.07 for the three‐component mixing model in the study region. The uncertainty terms for supra-permafrost water accounted for more than 50.0% of the total uncertainty, indicating that the $\delta$18O and $\delta$D variations of supra-permafrost water accounted for the majority of the uncertainty. Although there is some

uncertainty for hydrograph separation, isotope-based hydrograph separations are still valuable tools for evaluating the contribution of meltwater to water resources, and they are particularly helpful for improving our understanding of hydrological processes in cold regions, where there is a lack of observational data. "

Minor comments: Too many grammatical and word errors, as well as mistakes in graphs and captions. Authors should check their manuscript very carefully and ask for some native speaker to edit to make paper readable before submission. Thank you very much for your comments. Grammatical and word errors have been revised by native speaker. Other mistakes had been checked and revised.

Please also note the supplement to this comment:
https://www.hydrol-earth-syst-sci-discuss.net/hess-2019-530/hess-2019-530-AC3-supplement.pdf